# GENERALIZING TO NEW DYNAMICAL SYSTEMS VIA FREQUENCY DOMAIN ADAPTATION

## ABSTRACT

Learning the underlying dynamics from data with deep neural networks has shown remarkable potential in modeling various complex physical dynamics. However, current approaches are constrained in their ability to make reliable predictions in a specific domain and struggle with generalizing to unseen systems that are governed by the same general dynamics but differ in environmental characteristics. In this work, we formulate a parameter-efficient method, FNSDA, that can readily generalize to new dynamics via adaptation in the Fourier space. Specifically, FNSDA identifies the shareable dynamics based on the known environments using an automatic partition in Fourier modes and learns to adjust the modes specific for each new environment by conditioning on low-dimensional latent systematic parameters for efficient generalization. We experimentally evaluate FNSDA on representative families of nonlinear dynamics. The results show that FNSDA can achieve superior or competitive generalization performance compared to existing methods with a significantly reduced parameter cost.

## 1 INTRODUCTION

Standing at the intersection of deep learning and physics, we have witnessed tremendous progress being made in modeling complex natural phenomena from data directly (Ling et al., 2016; Brunton et al., 2016; Raissi et al., 2020). Successful and potential applications cover a broad spectrum of fields such as fluid dynamics (Kochkov et al., 2021; Ummenhofer et al., 2020), weather forecasting (Weyn et al., 2019; Pathak et al., 2022), astrophysics (Villanueva-Domingo et al., 2022) and biology (Aliee et al., 2022). Compared to traditional physical approaches endeavoring to build accurate numerical simulations, learned physical simulators with neural networks exhibit several desirable characteristics: less reliance on domain expertise in method designing, robustness to partially interpreted dynamics and incomplete physical models, and the capacity to offer solutions when dealing with high-dimensional data, making it a promising direction for advancing simulation capabilities and enabling more efficient and accurate modeling of complex systems (Wang & Yu, 2021).

Despite these compelling merits, deep learning approaches are notorious for their heavy dependency on large datasets for parameter learning and poor generalization performance when deployed in unseen environments with distinct characteristics (Wang & Yu, 2021). In contrast, numerical simulators can easily generalize to new dynamical systems providing specific environmental parameters (*e.g.*, external forces, initial values, boundary conditions). This disparity in generalization ability greatly impedes the widespread application of neural learned simulators due to the constant flux of real-world conditions. Consider, for one instance, in fluid flows simulation (Wang et al., 2022b), even though fluid flows are governed by the same equations, variations in buoyant forces necessitate separate deep learning models for accurate prediction. For another instance, in cardiac electrophysiology (Neic et al., 2017), inconsistencies in patients' body conditions can significantly impact the prediction of heart electrical behavior. Hence, there is a critical need for the development of deep learning models that can not only learn effectively and predict the dynamics of complex systems accurately, but also generalize well across heterogeneous domains.

This work embarks upon the generalization problem for neural learned simulators across different dynamical systems. To be more precise, we consider a problem setup where trajectories collected from several known environments are available for model training, and the model is expected to generalize to new environments with distinct environmental parameters based on a few observa-

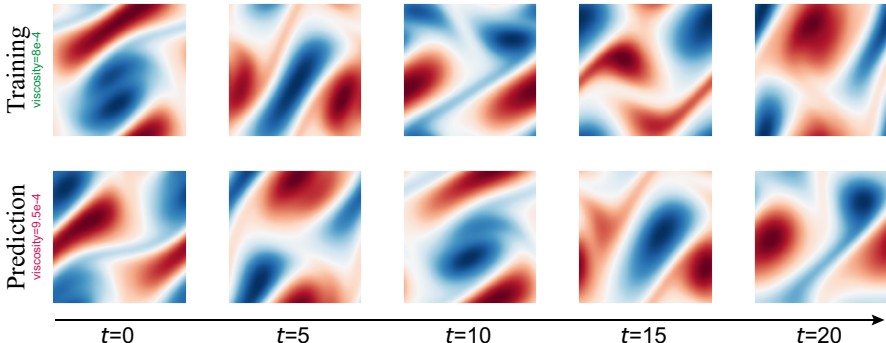

Figure 1: Dynamic forecast on Navier-Stokes equations. The learned simulator needs to generalize to new environments characterized by distinct viscosity.

tions. An example setup with the dynamics dictated by Navier-Stokes equations is shown in Fig. 1. This actually fits the scope of out-of-distribution generalization research that settles down to learning a model robust to distribution shift via meta-learning, disentanglement, or data manipulation (Wang et al., 2022a), and existing a few works learn such a shareable model of dynamical systems following the learning paradigms of meta-learning and feature disentanglement (Yin et al., 2021; Wang et al., 2022b; Kirchmeyer et al., 2022; Park et al., 2023; Jiang et al., 2023). Although these methods present some promising results on established benchmarks, they lack efficiency during adaptation as they require updating a large amount of parameters in the neural network either through gradient-of-gradient optimization caused by meta-learning, or conduct feature disentanglement based on multiple neural networks, which significantly prohibit their applications on resource-constrained edge devices (Yang et al., 2022; Liu et al., 2023).

To alleviate this, we propose *Fourier Neural Simulator for Dynamical Adaptation* (FNSDA), a parameter-efficient learning method that characterizes the behavior of complex dynamical systems in the frequency domain for rapid generalization towards new environments. Our work is inspired by the fact that changes in environmental parameters persistently affect both local and global dynamics, and such changes can be modeled by learning the Fourier representations in corresponding high and low modes (Cooley & Tukey, 1965; Van Loan, 1992). In addition, the complex non-linear relationship in the original temporal space can be converted into a linear relationship in the Fourier space, the difficulty of modeling is thus reduced (Orfanidis, 1995). Therefore, FNSDA builds its method in the Fourier domain. After performing Fourier transform on the input signals, FNSDA identifies the dynamics via a learnable filter that separates the Fourier modes into accounting for the commonalities and discrepancies among dynamic systems and learns their features through two independent weight multiplication. Based on this, FNSDA solely modifies linear dimensions for discrepancies in new systems, facilitating significantly reduced parameter cost and rapid speed of adaptation. Furthermore, when coupled with Swish activation, and training techniques such as regularization and cosine annealing learning rate scheduler, FNSDA exhibits a strong fitting capability for complex dynamics. We empirically evaluate FNSDA on two adaptation setups over four representative nonlinear dynamics, including ODEs with the Lotka-Volterra predator-prey interactions and the yeast glycolytic oscillation dynamics, PDEs derived from the Gray-Scott reaction-diffusion model and the more challenging incompressible Navier-Stokes equations. Our approach consistently achieves superior or competitive accuracy results compared to state-of-the-art approaches while requiring significantly fewer parameters to be updated during adaptation. In summary, we make the following three key contributions:

- We propose FNSDA, a novel method that embarks on the frequency domain for tackling the generalization challenge in modeling physical systems using neural network surrogates.

- We introduce a Fourier representation learning technique to characterize the commonalities and discrepancies among dynamical environments, yielding a largely reduced model complexity for rapid generalization.

- We provide empirical results to show that FNSDA outperforms or is competitive to other baseline methods on two evaluation tasks across various dynamics.

## 2  RELATED WORKS

**Out-of-Distribution Generalization.** The issue of out-of-distribution (OOD) generalization has emerged as a significant concern in machine learning. The primary objective is to learn robust models capable of generalizing effectively towards unseen environments, wherein the data may differ significantly from the training data. Existing methods commonly rely on multiple visible environments to acquire generalization capability, and we can categorize them into three categories according to their learning strategies. The first type is domain-invariant learning, which aims to learn a shareable feature space via robust optimization (Sagawa et al., 2020; Duchi et al., 2021), invariant risk minimization (Rosenfeld et al., 2021; Krueger et al., 2021) or disentanglement (Peng et al., 2019; Ilse et al., 2020). The second type is meta-learning based approaches, which employ the model-agnostic training procedure to mimic the train/test shift for better generalization (Li et al., 2018; Balaji et al., 2018; Dou et al., 2019). The last type is data manipulation which perturbs the original data and features to stimulate the unseen environments (Volpi et al., 2018; Yue et al., 2019; Zhou et al., 2021). A comprehensive review can refer to Wang et al. (2022a).

While tremendous progress is being achieved in this field, the proposed approaches typically confine themselves to a static configuration, thereby cannot adapt to our problem. Recently, some works have been devoted to generalization in continuously evolving environments (Nasery et al., 2021; Qin et al., 2022). Nonetheless, these methods require massive data to extract dynamic patterns and fail to extrapolate to novel environments that have not been seen during the training phase.

**Learning dynamical systems.** Deep learning models have recently gained considerable attention for simulating complex dynamics due to their ability to tackle complex, high-dimensional data (Chen et al., 2018; Wang et al., 2020; Sanchez-Gonzalez et al., 2020; Ummenhofer et al., 2020; Pfaff et al., 2021). While the predominant direction in contemporary research endeavors to incorporate inductive biases from physical systems, we aim to investigate the generalization to novel dynamical systems wherein changing is an intrinsic property and arise from various factors. Thus far, only a few works have considered this problem in dynamical systems. LEADS (Yin et al., 2021) presents a training strategy that learns to decouple commonalities and discrepancies between environments. DyAd (Wang et al., 2022b) follows a meta-learning style and adapts the dynamics model to unseen environments by decoding a time-invariant context. CoDA (Kirchmeyer et al., 2022) learns to condition the dynamics model on environment-specific and low-dimensional contextual parameters thus facilitating fast adaptation. FOCA (Park et al., 2023) also proceeds from a meta-learning manner but utilizes an exponential moving average trick to avoid second-order derivatives. Differing from these approaches to learning the environment-specific context on the temporal domain, we take a nuanced characterization in the frequency domain, this facilitates the modeling of dynamics in a linear manner and rapid adaptation.

**Fourier Transform.** Fourier transform is a mathematical tool that has significantly contributed to the evolution of deep learning techniques due to the efficiency of performing convolution (Bengio et al., 2007) and the capability of capturing long-range dependency (Zhang et al., 2018). It has the property that convolution in the time domain is equivalent to multiplication in the frequency domain. As a result, some works propose to incorporate Fourier transforms into neural network architectures to accelerate convolution computation (Mathieu et al., 2014; Lee-Thorp et al., 2022) and calculate the auto-correlation function efficiently (Sitzmann et al., 2020; Wen et al., 2021). In recent years, Fourier transform has also been combined with deep neural networks for solving various differential equations since it can transform differentiation into linear multiplication within the frequency domain (Li et al., 2021; 2022; Tran et al., 2023). More generally, it has been demonstrated the universal approximation property for learning the solution function (Kovachki et al., 2021). Building upon these seminal works, we propose a generalizable neural simulator that explicitly captures dynamic patterns by different modes within the Fourier space such that it can efficiently adapt to new physical environments by adjusting the coefficients of these modes.

## 3  METHODOLOGY

### 3.1  PROBLEM DEFINITION

We consider the problem of predicting the dynamics of complex physical systems (*e.g.*, fluid dynamics) with data collected from a set of environments $\mathcal{E}$. In particular, these systems are assumed to be

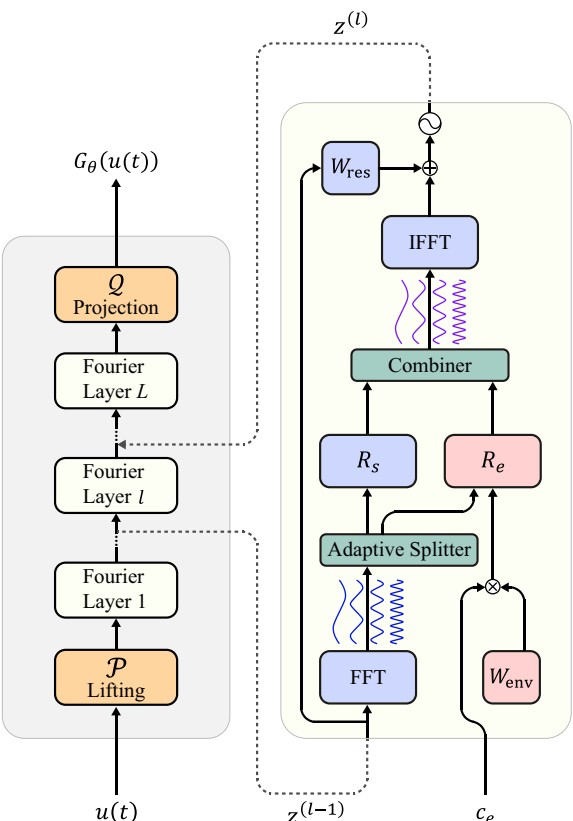

Figure 2: The architecture of FNSDA.

governed by the same family of nonlinear, coupled, differential equations, but their solutions differ due to *invisible* environment-specific parameterization. The general form of the system dynamics can be expressed as follows

$$\frac{\mathrm{d}\boldsymbol{u}}{\mathrm{d}t}(t) = F_e(\boldsymbol{u}(t)), \qquad t \in [0, T], \tag{1}$$

where $\boldsymbol{u}(t)$ are the time-dependent state variables taking their values from a bounded domain $\mathcal{U}$. The function $F_e$ usually is a non-linear operator lying in a functional vector field $\mathcal{F}$ and can vary in different environments due to some specific but unknown attributes (*e.g.*, physical parameters, external forces that affect the trajectories). When the spatial dependence is explicit and given, $\mathcal{U}$ becomes a $d'$-dimensional vector field over a bounded spatial domain $D \subset \mathbb{R}^{d'}$, and Eq. (1) corresponds to PDEs. In a similar vein, it corresponds to ODEs when $\mathcal{U} \subset \mathbb{R}^d$. In our experimental part, we consider both ODEs and PDEs.

In the generalization problem, we have access to several training environments $\mathcal{E}_{\mathrm{tr}} \subset \mathcal{E}$, where each environment $e \in \mathcal{E}_{\mathrm{tr}}$ is equipped with $N_{\mathrm{tr}}$ trajectories generated by the dynamical system defined in Eq. (1) with operator $F_e$. The goal is to learn a simulator $G_\theta$ parameterized by $\theta$ using the trajectories collected from $\mathcal{E}_{\mathrm{tr}}$, such that when provided with observations generated by an unknown $F_e$ in test environments $\mathcal{E}_{\mathrm{ev}} \subset \mathcal{E}$ (where $\mathcal{E}_{\mathrm{ev}} \cap \mathcal{E}_{\mathrm{tr}} = \emptyset$), $G_\theta$ can rapidly adapt and produce accurate predictions for these new environments. To evaluate the generalization capability of the learned simulator, we consider two adaptation tasks:

- **Inter-trajectory adaptation.** This task involves adapting the simulator $G_\theta$ to an unseen test environment $e \in \mathcal{E}_{\mathrm{ev}}$ using only one trajectory generated with $F_e$ over the time period $[0, T]$ for parameter updating. After that, $G_\theta$ needs to predict the dynamics for $N_{\mathrm{ev}}$ additional trajectories over $[0, T]$ by providing their initial states. This task emphasizes the rapid adaptation ability based on one-shot observation.

- **Extra-trajectory adaptation.** In this task, the simulator needs to produce precise predictions for $N_{\mathrm{ev}}$ trajectories for each test environment $e \in \mathcal{E}_{\mathrm{ev}}$. The front part of these trajectories can be used

for parameter adaptation ($[0, T_{\text{ad}}], T_{\text{ad}} < T$), and the model is required to predict the dynamics at subsequent time stamps ($t \in (T_{\text{ad}}, T]$). This task emphasizes the extrapolation ability towards the unseen future.

These two tasks encompass the typical usage scenarios of dynamical systems in the real world. In contrast to existing approaches that primarily focus on modeling the non-linear dynamics of diverse environments in the temporal domain, we turn to characterize the dynamics in the frequency domain, thus enabling rapid adaptation and accurate prediction for new systems.

## 3.2 FNSDA: FOURIER NEURAL SIMULATOR FOR DYNAMICAL ADAPTATION

In this work, we propose to tackle the generalization problem in modeling physical systems using neural network surrogates. Our designed method, FNSDA, learns a generalizable neural operator $G_\theta : \mathcal{U} \rightarrow \mathcal{U}$ with parameter $\theta$ as a surrogate model to approximate $F_e$ based on the trajectories collected from the environment $e$. This work is inspired by Fourier Neural Operator (FNO) (Li et al., 2021; Tran et al., 2023), which has shown promising results in modeling PDEs for a given dynamic. In the following sections, we will elaborate on how FNSDA acquires the fitting ability and generalization capability for new dynamical systems.

**Fourier Neural Operator.** This is an iterative approach first presented by Li et al. (2021) that learns the solution function for general PDEs represented by a kernel formulation. The overall computational flow of FNO for approximating the convolution operator is given as

$$G_\theta := \mathcal{Q} \circ \mathcal{L}^{(L)} \circ \cdots \circ \mathcal{L}^{(1)} \circ \mathcal{P}, \tag{2}$$

where $\circ$ represents function composition, $\mathcal{P}$ is the lifting operator that locally maps the input to a higher dimensional representation $\boldsymbol{z}^{(0)}$, $\mathcal{L}^{(l)}$ is the $l$-th non-linear operator layer $l \in \{1, ..., L\}$, and $\mathcal{Q}$ is the projection operator that locally maps the last latent representation $\boldsymbol{z}^{(L)}$ to the output. This iterative process is schematically depicted on the left part of Fig. 2.

A Fourier neural layer $\mathcal{L}^{(l)}$ in Eq. (2) is defined as follows

$$\mathcal{L}^{(l)} \left( \boldsymbol{z}^{(l)} \right) = \sigma^{(l)} \left( W_{\text{res}}^{(l)} \boldsymbol{z}^{(l)} + \mathcal{K}^{(l)}(\boldsymbol{z}^{(l)}) + b^{(l)} \right), \tag{3}$$

where $\mathcal{K}^{(l)}$ a kernel integral operator maps input to bounded linear output, $W_{\text{res}}^{(l)}$ a linear transformation, $b^{(l)}$ is a bias function and $\sigma^{(l)} : \mathbb{R} \rightarrow \mathbb{R}$ is a point-wise non-linear activation function. In particular, $\mathcal{K}^{(l)}$ is implemented by fast Fourier transform (Nussbaumer & Nussbaumer, 1981) with truncated modes as

$$\mathcal{K}^{(l)}(\boldsymbol{z}^{(l)}) = \text{IFFT}(R^{(l)} \cdot \text{FFT}(\boldsymbol{z}^{(l)})). \tag{4}$$

The Fourier-domain weight matrix $R^{(l)}$ is directly learned, and it yields $m^2$ parameters and computational complexity $\mathcal{O}(m^2 \hat{k}^d)$ for $m$-dimensional representation $\boldsymbol{z}^{(l)}$, $\hat{k}$ truncated Fourier modes, and $d$-dimensional problem. The overall computational complexity for a simulator with $L$ FNO layers is therefore $\mathcal{O}(Lm^2 \hat{k}^d)$. An essential characteristic making FNO outstand from conventional convolutional networks is that $\mathcal{P}$, $\mathcal{Q}$ and $\sigma$ are all defined as Nemitskiy operators, thus it can keep the functional attribute when input as a function (*e.g.*, the initial condition for a dynamical system).

**Improving generalization with FNSDA.** The vanilla FNO exhibits limitations in its ability to generalize across various dynamical systems due to the integration of all Fourier modes for modeling a specific domain. To acquire the generalization capability, FNSDA learns to partition the Fourier modes into two groups during the training phase, one accounting for the commonalities shared by different environments and the other for the discrepancies specific to each individual environment. After that, FNSDA only needs to adjust the parameters associated with modeling the discrepancies for the generalization to new environments while keeping other parameters fixed as already learned values. To facilitate rapid adaptation, we further introduce an efficient adjustment strategy with the usage of globally shared and low-dimensional systematical parameters.

In practice, $\text{FFT}(\boldsymbol{z}^{(l)})$ in Eq. (4) is implemented as a convolution on $\boldsymbol{z}^{(l)}$ with a function consisting of $\hat{k}$ Fourier modes caused by truncation, that is $\text{FFT}(\boldsymbol{z}^{(l)}) \in \mathbb{C}^{\hat{k} \times m}$. FNSDA separates these Fourier modes as follows

$$\begin{aligned} \text{FFT}_e(\boldsymbol{z}^{(l)}) &= \mathbf{K}^{(l)} \cdot \text{FFT}(\boldsymbol{z}^{(l)}) \\ \text{FFT}_s(\boldsymbol{z}^{(l)}) &= (\mathbf{1} - \mathbf{K}^{(l)}) \cdot \text{FFT}(\boldsymbol{z}^{(l)}), \end{aligned} \tag{5}$$

**Algorithm 1** Training for FNSDA

1: **Input:** Training environments $\mathcal{E}_{\text{tr}}$ each endowed with $N_{\text{tr}}$ trajectories; a simulator $G_\theta$ comprising of $L$ Fourier neural layers; environmental parameters $\boldsymbol{c}_e$; step size $\alpha$.
2: Randomly initialize $\theta$
3: Assign $\boldsymbol{c}_e \leftarrow \mathbf{0}$
4: **while** not converged **do**
5:     **for** each $e \in \mathcal{E}_{\text{tr}}$ **do**
6:         Compute the empirical loss $\mathcal{L}_{\text{data}}$ on $N_{\text{tr}}$ trajectories via Eq. (8);
7:     **end for**
8:     $\boldsymbol{c}_e \leftarrow \boldsymbol{c}_e - \alpha \nabla_{\boldsymbol{c}_e} \mathcal{L}_{\text{data}}$
9:     $\theta \leftarrow \theta - \eta \nabla_\theta \mathcal{L}_{\text{data}}$
10: **end while**

**Algorithm 2** Adaptation for FNSDA

1: **Input:** One test environment $e \in \mathcal{E}_{\text{ev}}$ with $N$ trajectories for adaptation; pretrained $G_\theta$; environmental parameters $\boldsymbol{c}_e$; activation parameters $\beta_e^{(l)}$; step size $\alpha$ and $\eta$.
2: Load pretrained $\theta$
3: Assign $\boldsymbol{c}_e \leftarrow \bar{\boldsymbol{c}}_{\text{tr}}, \beta_e^{(l)} \leftarrow \bar{\beta}_{\text{tr}}^{(l)}$
4: **while** not converged **do**
5:     Compute the empirical loss $\mathcal{L}_{\text{data}}$ on $N$ trajectories via Eq. (8);
6:     $\boldsymbol{c}_e \leftarrow \boldsymbol{c}_e - \alpha \nabla_{\boldsymbol{c}_e} \mathcal{L}_{\text{data}}$
7:     **for** $l = 1, ..., L$ **do**
8:         $\beta_e^{(l)} \leftarrow \beta_e^{(l)} - \eta \nabla_{\beta_e^{(l)}} \mathcal{L}_{\text{data}}$
9:     **end for**
10: **end while**

where $\mathbf{K}^{(l)} \in \mathbb{C}^{\hat{k}}$ is a learnable filter. As such, our method can automatically select appropriate modes to be kept or adjusted, which is an important property for its performance. Following the partition, the weight matrix $R^{(l)}$ is factorized into a combination of two independent matrices $R_e^{(l)} \in \mathbb{C}^{\hat{k} \times m}$ and $R_s^{(l)} \in \mathbb{C}^{\hat{k} \times m}$, each catering to the respective groups. Intuitively, $R_e^{(l)}$ ought to take different values for different systems, while directly treating it as a learnable metric would incur significant computational costs for adaptation as $Lm\hat{k}$ parameters would require updating when stacking $L$ FNO layers. To this end, we further introduce a resource-efficient strategy that achieves a similar adjustment effect by conditioning $R_e^{(l)}$ on low-dimensional and environment-specific systematic parameters $\boldsymbol{c}_e \in \mathbb{R}^{d_c}$, which can be given as

$$R_e^{(l)} = W_{\text{env}}^{(l)} \boldsymbol{c}_e, \quad \forall\, e \in \mathcal{E} \text{ and } l \in \{1, ..., L\}, \tag{6}$$

where $W_{\text{env}}^{(l)}$ is a learnable linear matrix. From Eq. (6) we notice that $\boldsymbol{c}_e$ is shared across all Fourier layers, this formulation effectively amplifies the impact of $\boldsymbol{c}_e$ on the behavior of $F_e$ such that we can only update the value of $\boldsymbol{c}_e$ when adapting to a new environment. In practice, $\boldsymbol{c}_e$ can be learned from the provided trajectories, we further apply them as conditional input for all Fourier layers.

Overall, Eq. (4) for FNSDA can be reformulated as

$$\mathcal{K}^{(l)}(\boldsymbol{z}^{(l)}) = \text{IFFT}\big(R_e^{(l)} \cdot \text{FFT}_e(\boldsymbol{z}^{(l)}) + R_s^{(l)} \cdot \text{FFT}_s(\boldsymbol{z}^{(l)})\big). \tag{7}$$

Compared to the vanilla FNO, FNSDA reorganizes the Fourier modes and conditions some of them on newly introduced systematical parameters $\boldsymbol{c}_e$. These seemingly minor modifications endow FNO a strong generalization ability due to (1) the preservation of its representation capability across all modes, without any degradation, and (2) the magnified impact of the vector $\boldsymbol{c}_e$ through the utilization of a hierarchical structure. Moreover, different from conventional approaches that employ complex training pipelines and additional networks to tackle the generalization issue directly in the temporal domain (Yin et al., 2021; Kirchmeyer et al., 2022; Park et al., 2023), our frequency domain-based method benefits from the reduced difficulty in approximating the non-linear dynamics and inferring the value of $\boldsymbol{c}_e$ from trajectories. We further show that when incorporated with Swish activation function and training techniques like regularization and the cosine annealing learning rate scheduler, our FNSDA exhibits a powerful generalization capability across various dynamical systems and fitting ability for seen dynamics no matter for PDEs or ODEs.

### 3.3 IMPLEMENTATION

**Swish activation.** We choose Swish activation (Ramachandran et al., 2018) as the activation function in Eq. (3) due to its superior ability in a variety of tasks. It is a smooth non-monotonic function with a learnable parameter that takes the form $\sigma^{(l)}(\boldsymbol{x}) = \boldsymbol{x} \cdot \text{sigmoid}(\beta_e^{(l)} \boldsymbol{x})$, where $\boldsymbol{x}$ represents the provided intermediate representations and $\beta_e^{(l)}$ is a learnable parameter for the $l$-th layer. Swish activation brings non-linearity into the network such that our neural network surrogate can capture the complex interactions between the input features. To tailor it to the multi-environment dynamics forecasting scenario, we maintain a distinct $\beta_e^{(l)}$ for each individual environment.

**Model training and adaptation.** In real-world applications, systematical parameters tend to take similar values across different systems, while small changes in their values can have a substantial impact on the dynamics, especially for long-range prediction (Sanchez-Gonzalez et al., 2020; Han et al., 2022). Therefore, we introduce regularization to impose constraints on the behavior of these systematical parameters. Specifically, providing $N$ trajectories collected from a single environment $e \in \mathcal{E}$, we optimize our model to minimize a unified empirical data loss as

$$\mathcal{L}_{\text{data}}(G_\theta, F_e) = \frac{1}{N} \sum_{j=1}^{N} \int_0^T ||F_e(\boldsymbol{u}_j(t)) - G_\theta(\boldsymbol{u}_j(t); \boldsymbol{c}_e)||_2^2 \, \mathrm{d}t + \lambda ||\boldsymbol{c}_e||_2^2. \tag{8}$$

At the training stage, when optimizing our model to minimize the loss Eq. (8) for all dynamics forecasting tasks in training environments $\mathcal{E}_{\text{tr}}$, the general dynamic is effectively learned and inherent in its parameters $\theta$. As a result, when adapting to a previously unseen environment $e \in \mathcal{E}_{\text{ev}}$, we only need to update $\boldsymbol{c}_e$ and $\beta_e^{(l)}$ to minimize Eq. (8) based on newly collected trajectories, this makes our method quite efficient for practical applications. Furthermore, we initialize $\boldsymbol{c}_e$ and $\beta_e^{(l)}$ as the average of their learned values in the training environments to further speed up the adaptation process. The training and adaptation procedures are outlined in Algorithm 1 and 2, respectively.

## 4  Experiments

In this section, we evaluate FNSDA on four representative dynamical systems that have been widely employed by various fields *e.g.*, chemistry, biology and fluid dynamics. These systems all exhibit complex non-linearity in either temporal or spatiotemporal domains. We compare our method with other baselines on both inter-trajectory and extra-trajectory adaptation tasks.

### 4.1  Experimental Setup

**Datasets.** We experiment on two ODEs and two PDEs: (1) Lotka-Volterra (LV) (Lotka, 1925). This is an ODE dataset describing the dynamics of a prey-predator pair and their interaction within an ecosystem. The environmental parameters are the quantities of the prey and the predator, and we vary their values to imitate different dynamical systems. (2) Glycolitic-Oscillator (GO) (Daniels & Nemenman, 2015). An ODE dataset depicts yeast glycolysis oscillations for biochemical dynamics inference. We adjust the parameters of the glycolytic oscillators to generate different systems. (3) Gray-Scott (GS) Pearson (1993). A PDE dataset describes the spatiotemporal patterns of reaction-diffusion system. We vary the values of reaction parameters for each environment. (4) Navier-Stokes system (NS) (Stokes, 1851), a two-dimensional PDE dataset exhibiting complex spatiotemporal dynamics of incompressible flows. The environmental parameter is viscosity, we take different viscosity to mimic environmental change. For the LV and GO datasets, each training system is equipped with $N_{\text{tr}} = 100$ trajectories for parameter learning. The model is evaluated on $N_{\text{ev}} = 50$ trajectories from new systems. For the GS and NS datasets, we let $N_{\text{tr}} = 50$ and $N_{\text{ev}} = 50$ for training and evaluation, respectively. More details for dataset generation can be found in Appendix B.

**Baselines.** The methods for comparison include: (1) ERM (Vapnik, 1998); (2) ERM-adp, fine-tuning ERM learned parameters to adapt to new environments; (3) LEADS (Yin et al., 2021); (4) CoDA (Kirchmeyer et al., 2022), we use $\ell_1$ (CoDA-$\ell_1$) and $\ell_2$ norm (CoDA-$\ell_2$) for the regularization on the context and hypernetwork as suggested by Kirchmeyer et al. (2022); (5) FoCA (Park et al., 2023). We implement these methods following the neural network architecture presented in Kirchmeyer et al. (2022).

FNSDA is implemented in the PyTorch (Paszke et al., 2017) platform. For the experiments on LV and GO datasets, we use two Fourier layers with $\hat{k} = 10$ frequency modes. For the GS and NS datasets, we employ four Fourier layers with $\hat{k} = 12$ truncated modes. Across these datasets, the dimension of environmental parameter $\boldsymbol{c}_e$ is set as $d_c = 20$ and the coefficient for regularization as $\lambda = 1\text{e-4}$. To calculate the trajectory loss presented in Eq. (8), we employ numerical solvers to approximate the integral. Specifically, we utilize RK4 solver for the LV, GO and GS datasets, and Euler solver for the NS dataset. We optimize the model using Adam (Kingma & Ba, 2015) with an initial learning rate of 1e-3, and the learning rate is updated with a cosine annealing schedule. For simplicity, we set $\alpha$ equal to $\eta$. We find that cosine annealing schedule with warmup is effective for model training but failed for adaptation, so we only use warmup over the first 500 epochs when

Table 1: Inter-trajectory adaptation results. We measure the RMSE ($\times 10^{-2}$) and MAPE values per trajectory. Smaller is better ($\downarrow$). # Params indicate the number of updated parameters for adapting to new environments. Detailed results with standard deviations are available in Appendix C.

| Algorithm | LV | | | GO | | | GS | | | NS | | |
|---|---|---|---|---|---|---|---|---|---|---|---|---|
| | RMSE | MAPE | #Params | RMSE | MAPE | #Params | RMSE | MAPE | #Params | RMSE | MAPE | #Params |
| ERM | 48.310 | 3.081 | - | 18.688 | 0.355 | - | 8.120 | 3.370 | - | 5.906 | 0.416 | - |
| ERM-adp | 47.284 | 2.170 | 0.008M | 33.161 | 0.516 | 0.008M | 9.924 | 4.665 | 0.076M | 17.516 | 1.491 | 0.232M |
| LEADS | 69.604 | 2.440 | 0.043M | 33.782 | 0.688 | 0.043M | 23.017 | 2.185 | 0.020M | 36.855 | 0.974 | 1.162M |
| CoDA-$\ell_2$ | 4.674 | 0.554 | 0.035M | 46.461 | 0.688 | 0.035M | 20.017 | 12.007 | 0.381M | 2.784 | 0.299 | 0.465M |
| CoDA-$\ell_1$ | 5.044 | 0.636 | 0.035M | 46.051 | 0.729 | 0.035M | 28.465 | 6.001 | 0.381M | **2.773** | **0.297** | 0.465M |
| FOCA | 21.321 | 0.601 | 0.013M | 44.020 | 0.618 | 0.013M | 14.678 | 4.565 | 0.028M | 17.115 | 1.854 | 0.237M |
| FNSDA | **3.736** | **0.216** | 0.088K | **8.541** | **0.229** | 0.088K | **2.700** | **0.826** | 0.096K | 3.625 | 0.355 | 0.096K |

Table 2: Extra-trajectory adaptation results. We measure the RMSE ($\times 10^{-2}$) and MAPE per trajectory. Smaller is better ($\downarrow$). Detailed results with standard deviations are available in Appendix C.

| Algorithm | LV | | GO | | GS | | NS | |
|---|---|---|---|---|---|---|---|---|
| | RMSE | MAPE | RMSE | MAPE | RMSE | MAPE | RMSE | MAPE |
| ERM | 43.969 | 2.347 | 18.233 | 0.306 | 7.059 | 3.027 | 4.969 | 0.383 |
| ERM-adp | 95.193 | 3.465 | 23.522 | 0.566 | 163.670 | 83.508 | 31.521 | 5.746 |
| LEADS | 88.214 | 3.390 | 34.617 | 0.729 | 28.115 | 18.222 | 39.398 | 1.265 |
| CoDA-$\ell_2$ | **29.660** | 1.117 | 39.589 | 0.402 | 11.452 | 2.769 | **2.797** | **0.280** |
| CoDA-$\ell_1$ | 31.088 | 1.179 | 53.702 | 0.467 | 6.943 | **0.921** | 2.844 | 0.285 |
| FOCA | 77.046 | 6.725 | 76.194 | 1.484 | 49.476 | 35.736 | 11.238 | 1.131 |
| FNSDA | 33.774 | **0.420** | **14.918** | **0.236** | **5.011** | 2.695 | 3.823 | 0.370 |

training our model. We report the results in both the Root Mean Square Error (RMSE) and Mean Absolute Percentage Error (MAPE) for evaluation.

## 4.2 RESULTS

**Results of inter-trajectory adaptation.** The results in terms of inter-trajectory adaptation tasks are presented in Table 1. We also report the number of updated parameters during the adaptation procedure for each approach. As seen, FNSDA achieves the smallest forecast error on the LV, GO and GS datasets, exhibiting a noticeable improvement over other baselines. On the NS dataset, it performs also competitively, with results second only to CoDA. These results confirm the good generalization capability of our method. Furthermore, different from other methods requiring large amounts of parameters to be updated when adapting to a new environment, our FNSDA alleviates this dependence by requiring only a few updated parameters for adaptation. Such appealing property is actually attributed to the magnified impact of $\mathbf{c}_e$ stemming from the employed automatic partition strategy and hierarchical structure, prompting the practical usage of our method in resource-constrained and partial reconfigurable devices where updating all parameters is impractical (Vipin & Fahmy, 2018).

**Results of extra-trajectory adaptation.** The results for extra-trajectory adaptation tasks are shown in Table 2. FNSDA consistently obtains the best or at least competitive results across these experimental setups, demonstrating strong flexibility for various application scenarios. CoDA also exhibits promising performance, particularly on the NS dataset when utilizing $\ell_2$ norm. However, due to the existence of accumulation error (Sanchez-Gonzalez et al., 2020), most approaches exhibit higher forecast errors compared to the results obtained in the inter-trajectory adaptation task. This necessitates the development of specific methods or regularization techniques to mitigate this issue.

## 4.3 ABLATION STUDIES

**Effect of automatic partition strategy.** Fig. 3 displays the comparison results of FNSDA utilizing different Fourier modes splitting strategies for the inter-trajectory adaptation task on the LV dataset. Notably, FNSDA employing an automatic partition strategy demonstrates superior performance. A noticeable performance degradation can be observed when only updating low Fourier modes. This may be attributed to that environmental parameters own a preference for adjusting the high-frequency information of dynamics via small value changes, while modifying high modes only fails to yield optimal results. We further conducted a comparison by employing alternative splitting with

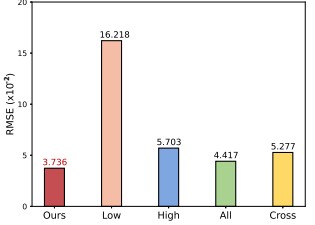

Figure 3: Competitions of different partition strategies.

Table 3: Comparisons of cross partition with different ratios. We report the RMSE $(\times 10^{-2})$ results.

| Split ratio | 4:1 | 3:2 | 1:1 | 2:3 | 1:4 |
|---|---|---|---|---|---|
| Inter-trajectory | 18.055 | 5.716 | 5.277 | 9.074 | 5.619 |
| Extra-trajectory | 50.304 | 60.574 | 65.126 | 41.219 | 67.216 |

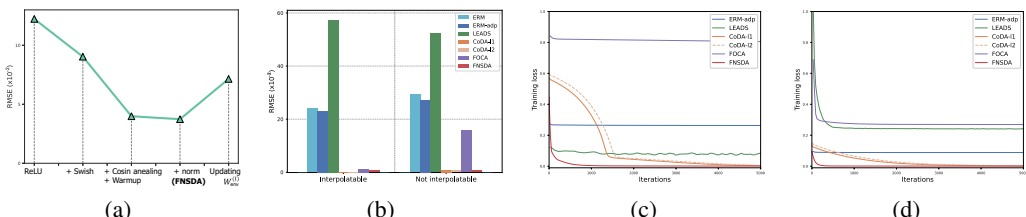

|  |  |  |  |
|---|---|---|---|
| (a) | (b) | (c) | (d) |

Figure 4: (a) training techniques, (b) different distribution discrepancy, (c) the convergence curves for inter-trajectory adaptation, (d) the convergence curves for extra-trajectory adaptation.

various ratios that can keep both the high and low modes within each group. The results, as presented in Table 3, indicate that these manual partition strategies can not lead to desired performance.

**Ablation on training techniques.** We performed an ablation study of employed training techniques to show the necessity of each of them, and the results on the LV dataset can be found in Fig. 4 (a). We start with a plain model using ReLU activation, which yielded an error of $12.2$. After replacing it with Swish, the error decreases by $3.2$. Moreover, incorporating cosine annealing scheduler and warmup further decreased the error by $5.0$. The addition of regularization on $c_e$ resulted in an additional error reduction of $0.3$. However, making $W_{\text{env}}^{(l)}$ tunable did not improve the performance due to the existence of overfitting in one trajectory adaptation.

**Analysis on distribution discrepancy.** To assess the performance of our method in handling various distribution shifts, we created two test environments on the LV dataset that exhibit different levels of distribution discrepancy from the training environments: one with environmental parameters can be interpolated from training environments and the other not. The experiment results are illustrated in Fig. 4 (b). Most methods present a degradation in performance when adapting to the test environment with environmental parameters that are not interpolatable. Conversely, our method still achieves a low forecast error in this challenging scenario, indicating its robustness and effectiveness in handling such distribution shifts.

**Analysis on Adaptation efficiency.** We further investigate the convergence speed of FNSDA for adapting to new environments. Specifically, we visualize the forecast error with respect to iteration steps during inter-trajectory and extra-trajectory adaptation processes on the LV dataset in Fig. 4 (c) and (d), respectively. FNSDA exhibits an appealing rapid adaptation capability to new environments. In the inter-trajectory adaptation task, it is able to converge to a stable value within 1,200 iterations. For the extra-trajectory adaptation task, FNSDA achieves convergence in a mere 100 iterations. Furthermore, unlike CoDA requires maintaining a duplicate model for updating all parameters, FNSDA eliminates this dependency, making it applicable in some resource-constrained edge devices.

## 5  CONCLUSION

In this paper, we propose a novel approach, FNSDA, to deal with the generalization problem in neural learned simulators for complex dynamical systems. By capitalizing on the frequency domain, FNSDA effectively captures the commonalities and discrepancies among various dynamical environments, which leads to a simplified model structure and expedited adaptation process. Comprehensive evaluations on two adaptation tasks across a diverse set of datasets demonstrate the superiority or competitiveness of FNSDA in comparison to existing methods, along with its remarkable speed in adapting to new environments.

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

**Table of Contents**

# A  LIMITATIONS AND FUTURE WORKS

## A.1  LIMITATIONS

**Requirements on high-quality data.** As a data-driven method, FNSDA relies on the quality and quantity of data. In current benchmark datasets, the training data for FNSDA are generated using well-designed numerical simulators. However, in practical applications, we may not have access to such accurate simulators and may need to learn a simulator from noisy or corrupted observations directly. Such limitations have the potential to impact the method's generalization capability negatively.

**Application to larger systems.** We primarily focus on relatively small dynamical systems governed by differential equations in this study. The scalability of FNSDA to much larger and more complex systems, such as those encountered in climate modeling or large-scale biological networks, remains an open question. The efficiency and generalization capabilities of FNSDA may be affected when dealing with such large-scale problems.

## A.2  FUTURE WORKS

**Accelerating Fourier transform.** Each Fourier layer in $G_\theta$ includes one Fourier transform and inverse Fourier transform, these two time-consuming operations actually hinder the training of FNSDA. To alleviate this, one may consider some truncation techniques for spectrum Poli et al. (2022); Tran et al. (2023), and reducing the number of performing transforms in architecture design Poli et al. (2022).

**Incorporating physical constraints and prior knowledge.** Incorporating physical constraints or prior knowledge into the FNSDA framework could lead to more robust and accurate predictions across a wider range of dynamical systems Wang & Yu (2021); Hansen et al. (2023). This could involve developing methods to fuse the learned representations with existing physical models, or designing novel architectures that explicitly enforce the satisfaction of physical constraints during the learning process.

**Extending to other types of dynamical systems.** Although FNSDA leverages the Fourier transform to linearize the relationships within the input signals, it is uncertain how well the method would perform on highly nonlinear or chaotic systems Sanchez-Gonzalez et al. (2020); Li et al. (2022). These systems may present additional challenges in modeling, generalization, and adaptation that have not been fully addressed in the current work. Exploring the performance of FNSDA on such systems would be a valuable direction for future research.

# B  EXPERIMENTAL SETTINGS

In this section, we present a comprehensive overview of the equations governing all the dynamical systems considered in the work. In addition, we will also delve into the specificities of data generation that are unique to each of these systems.

**Lotka-Volterra (LV).** This classical model is utilized to elucidate the dynamics underlying the interaction between a predator and its prey. Specifically, the governing equations are described by a system of ODE:

$$\frac{\mathrm{d}x}{\mathrm{d}t} = \alpha x - \beta xy$$

$$\frac{\mathrm{d}y}{\mathrm{d}t} = \delta x - \gamma xy$$

where $x, y$ are variables respectively indicate the quantity of the prey and the predator and $\alpha, \beta, \gamma, \delta$ are parameters defining the interaction process between the two species.

For model training, we consider 9 systems $\mathcal{E}_{\text{tr}}$ with parameters $\beta, \delta \in \{0.5, 0.75, 1.0\}^2$. And for evaluation, we consider 4 systems $\mathcal{E}_{\text{ev}}$ with parameters $\beta, \delta \in \{0.625, 1.125\}^2$. We maintain a constant value of $\alpha = 0.5$ and $\gamma = 0.5$ across all environments. Each of the training environments is equipped with $N_{\text{tr}} = 100$ trajectories, while each test environment is equipped with $N_{\text{ev}} = 50$ trajectories. Besides, all these trajectories use initial conditions randomly sampled from a uniform distribution $\text{Unif}([1, 3]^2)$ and evolve on a temporal grid with $\Delta t = 0.5$ and temporal horizon $T = 20$. Furthermore, for extra-trajectory prediction tasks on the LV dataset, we let $T_{\text{ad}} = 5$ for adaptation purposes, and models are expected to predict 15 seconds of future states.

**Glycolitic-Oscillator (GO).** The glycolytic oscillators refer to a mathematical model that characterizes the dynamics of yeast glycolysis following the ODE:

$$\frac{dS_1}{dt} = J_0 - \frac{k_1 S_1 S_6}{1 + (1/K_1^q)S_6^q}$$

$$\frac{dS_2}{dt} = 2\frac{k_1 S_1 S_6}{1 + (1/K_1^q)S_6^q} - k_2 S_2(N - S_5) - k_6 S_2 S_5$$

$$\frac{dS_3}{dt} = k_2 S_2(N - S_5) - k_3 S_3(A - S_6)$$

$$\frac{dS_4}{dt} = k_3 S_3(A - S_6) - k_4 S_4 S_5 - \kappa(S_4 - S_7)$$

$$\frac{dS_5}{dt} = k_2 S_2(N - S_5) - k_4 S_4 S_5 - k_6 S_2 S_5$$

$$\frac{dS_6}{dt} = -2\frac{k_1 S_1 S_6}{1 + (1/K_1^q)S_6^q} + 2k_3 S_3(A - S_6) - K_5 S_6$$

$$\frac{dS_7}{dt} = \psi\kappa(S_4 - S_7) - k S_7$$

where $S_1, S_2, S_3, S_4, S_5, S_6, S_7$ (states) denote the concentrations of 7 biochemical species and $J_0, k_1, k_2, k_3, k_4, k_5, k_6, K_1, q, N, A, \kappa, \psi$ and $k$ are the parameters determining the behavior of the glycolytic oscillators.

For training data generation, we consider 9 systems $\mathcal{E}_{\text{tr}}$ with parameters $k_1 \in \{100, 90, 80\}$, $K_1 \in \{1, 0.75, 0.5\}$. And for evaluation, we consider 4 systems $\mathcal{E}_{\text{ev}}$ with parameters $k_1 \in \{85, 95\}$, $K_1 \in \{0.625, 0.875\}$. We fix the parameters $J_0 = 2.5$, $k_2 = 6$, $k_3 = 16$, $k_4 = 100$, $k_5 = 1.28$, $q = 4$, $N = 1$, $A = 4$, $\kappa = 13$, $\psi = 0.1$ and $k = 1.8$ across all environments. For the GO dataset, each training environment is equipped with $N_{\text{tr}} = 100$ trajectories, and each test environment is equipped with $N_{\text{ev}} = 50$ trajectories. The trajectories are generated using initial conditions drawn from a uniform distribution as outlined in Kirchmeyer et al. (2022) and evolving on a temporal grid with $\Delta t = 0.05$ second and temporal horizon $T = 2$ seconds. Notably, for the extra-trajectory prediction tasks, all models are provided with the first $T_{\text{ad}} = 0.5$ seconds of observations for adaptation purposes, after which they are expected to predict the subsequent 1.5 seconds of future states.

**Gray-Scott (GS).** This is a 2d PDE dataset comprising the data for a reaction-diffusion system with complex spatiotemporal patterns derived from the following PDE equation:

$$\frac{\partial u}{\partial t} = D_u \Delta u - uv^2 + F(1 - u)$$

$$\frac{\partial v}{\partial t} = D_v \Delta v - uv^2 + (F + k)v$$

where $u$ and $v$ are the concentrations of two chemical components taking value in the spatial domain $S$ with periodic boundary conditions. $D_u$ is the diffusion coefficient for $u$, and $D_v$ is the diffusion coefficient for $v$. $F$ and $k$ denote the reaction parameters for this system.

For training data generation, we create 4 training environments $\mathcal{E}_{\text{tr}}$ via varying the reaction parameters $F \in \{0.30, 0.39\}$, $k \in \{0.058, 0.062\}$. While for evaluation, we generate 4 test environments $\mathcal{E}_{\text{ev}}$ with parameters $F \in \{0.33, 0.36\}$, $k \in \{0.59, 0.61\}$. Across these environments, we keep

Table 4: Inter-trajectory adaptation results. We measure the RMSE $(\times 10^{-2})$ and MAPE values per trajectory. Smaller is better $(\downarrow)$. # Params indicate the number of updated parameters for adapting to new environments.

| Algorithm | LV | | GO | | GS | | NS | |
|---|---|---|---|---|---|---|---|---|
| | RMSE | MAPE | RMSE | MAPE | RMSE | MAPE | RMSE | MAPE |
| ERM | $48.310_{\pm18.243}$ | $3.081_{\pm5.015}$ | $18.688_{\pm0.378}$ | $0.355_{\pm0.072}$ | $8.120_{\pm0.815}$ | $3.370_{\pm2.882}$ | $5.906_{\pm1.833}$ | $0.416_{\pm0.389}$ |
| ERM-adp | $47.284_{\pm9.373}$ | $2.170_{\pm2.227}$ | $33.161_{\pm1.115}$ | $0.516_{\pm0.214}$ | $9.924_{\pm1.617}$ | $4.665_{\pm3.327}$ | $17.516_{\pm4.866}$ | $1.491_{\pm9.865}$ |
| LEADS | $69.604_{\pm22.670}$ | $2.440_{\pm4.278}$ | $33.782_{\pm1.197}$ | $0.688_{\pm0.148}$ | $23.017_{\pm0.052}$ | $2.185_{\pm2.941}$ | $36.855_{\pm1.748}$ | $0.974_{\pm2.055}$ |
| CoDA-$\ell_2$ | $4.674_{\pm2.563}$ | $0.554_{\pm0.631}$ | $46.461_{\pm1.964}$ | $0.688_{\pm0.186}$ | $20.017_{\pm1.117}$ | $12.007_{\pm9.687}$ | $2.784_{\pm0.862}$ | $0.299_{\pm0.581}$ |
| CoDA-$\ell_1$ | $5.044_{\pm2.817}$ | $0.636_{\pm0.737}$ | $46.051_{\pm1.661}$ | $0.729_{\pm0.204}$ | $28.465_{\pm2.484}$ | $6.001_{\pm4.366}$ | $\mathbf{2.773}_{\pm0.845}$ | $\mathbf{0.297}_{\pm0.565}$ |
| FOCA | $21.321_{\pm18.243}$ | $0.601_{\pm0.590}$ | $44.020_{\pm1.133}$ | $0.618_{\pm0.309}$ | $14.678_{\pm1.175}$ | $4.565_{\pm3.534}$ | $17.115_{\pm5.780}$ | $1.854_{\pm6.513}$ |
| FNSDA | $\mathbf{3.736}_{\pm2.348}$ | $\mathbf{0.216}_{\pm0.221}$ | $\mathbf{8.541}_{\pm0.172}$ | $\mathbf{0.229}_{\pm0.076}$ | $\mathbf{2.700}_{\pm0.394}$ | $\mathbf{0.826}_{\pm0.500}$ | $3.625_{\pm0.882}$ | $0.355_{\pm0.579}$ |

Table 5: Extra-trajectory adaptation results. We measure the RMSE $(\times 10^{-2})$ and MAPE values per trajectory. Smaller is better $(\downarrow)$.

| Algorithm | LV | | GO | | GS | | NS | |
|---|---|---|---|---|---|---|---|---|
| | RMSE | MAPE | RMSE | MAPE | RMSE | MAPE | RMSE | MAPE |
| ERM | $43.969_{\pm22.576}$ | $2.347_{\pm5.121}$ | $18.233_{\pm0.685}$ | $0.306_{\pm0.096}$ | $7.059_{\pm1.198}$ | $3.027_{\pm3.712}$ | $4.969_{\pm1.333}$ | $0.383_{\pm0.428}$ |
| ERM-adp | $95.193_{\pm15.477}$ | $3.465_{\pm2.805}$ | $23.522_{\pm1.599}$ | $0.566_{\pm0.162}$ | $163.670_{\pm39.819}$ | $83.508_{\pm118.951}$ | $31.521_{\pm2.070}$ | $5.746_{\pm6.742}$ |
| LEADS | $88.214_{\pm28.864}$ | $3.390_{\pm3.602}$ | $34.617_{\pm1.650}$ | $0.729_{\pm0.210}$ | $28.115_{\pm1.528}$ | $18.222_{\pm21.688}$ | $39.398_{\pm2.038}$ | $1.265_{\pm1.676}$ |
| CoDA-$\ell_2$ | $\mathbf{29.660}_{\pm27.787}$ | $1.117_{\pm2.609}$ | $39.589_{\pm1.646}$ | $0.402_{\pm0.370}$ | $11.452_{\pm2.496}$ | $2.769_{\pm3.397}$ | $\mathbf{2.797}_{\pm0.769}$ | $\mathbf{0.280}_{\pm0.669}$ |
| CoDA-$\ell_1$ | $31.088_{\pm28.311}$ | $1.179_{\pm2.677}$ | $53.702_{\pm5.216}$ | $0.467_{\pm0.349}$ | $6.943_{\pm2.161}$ | $\mathbf{0.921}_{\pm1.398}$ | $2.844_{\pm0.746}$ | $0.285_{\pm0.683}$ |
| FOCA | $77.046_{\pm13.368}$ | $6.725_{\pm0.853}$ | $76.194_{\pm2.778}$ | $1.484_{\pm0.401}$ | $49.476_{\pm6.062}$ | $35.736_{\pm47.329}$ | $11.238_{\pm2.058}$ | $1.131_{\pm3.302}$ |
| FNSDA | $33.774_{\pm28.122}$ | $\mathbf{0.420}_{\pm0.467}$ | $\mathbf{14.918}_{\pm0.861}$ | $\mathbf{0.236}_{\pm0.079}$ | $\mathbf{5.011}_{\pm1.967}$ | $2.695_{\pm3.288}$ | $3.823_{\pm0.997}$ | $0.370_{\pm0.614}$ |

the diffusion coefficients fixed as $D_u = 0.2097$ and $D_v = 0.105$. The space is discretized on a 2D grid with dimension $32 \times 32$ and spatial resolution $\Delta s = 2$ following the setup in Kirchmeyer et al. (2022). For each training and test environment, we sample $N_{\text{tr}} = N_{\text{ev}} = 50$ initial conditions uniformly from three two-by-two squares in $S$ to generate the trajectories on a temporal grid with $\Delta t = 40$ second and temporal horizon $T = 400$ seconds. For the extra-trajectory prediction tasks, we set the visible time span as $T_{\text{ad}} = 80$ seconds for adaptation and the model needs to produce the prediction for the states in the following 320 seconds.

**Navier-Stokes (NS).** The Navier-Stokes equations are a set of PDEs that describe the dynamics of incompressible flows in a 2D space. These equations can be expressed in the form of a vorticity equation as follows:

$$
\begin{aligned}
\frac{\partial w}{\partial t} &= -v\nabla w + \nu\Delta w + f \\
\nabla v &= 0 \\
w &= \nabla \times v
\end{aligned}
$$

where $v$ denotes the velocity field and $w$ represents the vorticity, $\nu$ denotes the viscosity, and $f$ is a constant forcing term. The domain is subject to periodic boundary conditions.

For training data generation, we consider 5 systems $\mathcal{E}_{\text{tr}}$ with parameters $\nu \in \{8 \cdot 10^{-4}, 9 \cdot 10^{-4}, 1.0 \cdot 10^{-3}, 1.1 \cdot 10^{-3}, 1.2 \cdot 10^{-3}\}$. While for evaluation, we generate 4 systems $\mathcal{E}_{\text{ev}}$ with parameters $\nu \in \{8.5 \cdot 10^{-4}, 9.5 \cdot 10^{-4}, 1.05 \cdot 10^{-3}, 1.15 \cdot 10^{-3}\}$. The space is discretized on a 2D grid with dimension $32 \times 32$ and we set $f(x, y) = 0.1(\sin(2\pi(x + y)) + \cos(2\pi(x + y)))$, where $x, y$ are coordinates on the discretized domain following Kirchmeyer et al. (2022). For each training and test environment, we sample $N_{\text{tr}} = N_{\text{ev}} = 50$ initial conditions from the distribution described in Li et al. (2021) to generate the trajectories on a temporal grid with $\Delta t = 1$ second and temporal horizon $T = 10$ seconds. For the extra-trajectory prediction tasks on the NS dataset, all models are provided with the first $T_{\text{ad}} = 2$ seconds of observations for adaptation. Subsequently, they are expected to predict the states in the following 8 seconds.

Table 6: In-domain test results. We measure the RMSE ($\times 10^{-2}$) and MAPE values per trajectory. Smaller is better ($\downarrow$).

| Algorithm | LV | | GO | | GS | | NS | |
|---|---|---|---|---|---|---|---|---|
| | RMSE | MAPE | RMSE | MAPE | RMSE | MAPE | RMSE | MAPE |
| ERM | $39.753_{\pm 14.014}$ | $0.901_{\pm 1.052}$ | $23.946_{\pm 1.187}$ | $0.462_{\pm 0.170}$ | $18.491_{\pm 0.009}$ | $37.596_{\pm 2.882}$ | $6.793_{\pm 2.636}$ | $1.435_{\pm 11.744}$ |
| LEADS | $47.266_{\pm 12.590}$ | $0.940_{\pm 2.669}$ | $29.381_{\pm 0.975}$ | $0.698_{\pm 0.373}$ | $29.381_{\pm 0.975}$ | $\mathbf{0.698}_{\pm 2.941}$ | $36.551_{\pm 2.050}$ | $1.532_{\pm 9.687}$ |
| CoDA-$\ell_2$ | $4.591_{\pm 2.766}$ | $0.196_{\pm 0.590}$ | $5.567_{\pm 0.105}$ | $0.095_{\pm 0.061}$ | $5.254_{\pm 1.062}$ | $6.228_{\pm 9.687}$ | $\mathbf{2.813}_{\pm 0.932}$ | $\mathbf{0.660}_{\pm 5.333}$ |
| CoDA-$\ell_1$ | $3.947_{\pm 1.942}$ | $0.201_{\pm 0.634}$ | $\mathbf{5.400}_{\pm 0.094}$ | $\mathbf{0.091}_{\pm 0.057}$ | $6.260_{\pm 1.358}$ | $7.275_{\pm 4.366}$ | $3.521_{\pm 0.782}$ | $0.748_{\pm 5.235}$ |
| FOCA | $39.753_{\pm 14.014}$ | $0.901_{\pm 0.326}$ | $46.530_{\pm 1.938}$ | $2.522_{\pm 3.591}$ | $46.530_{\pm 1.938}$ | $0.737_{\pm 3.534}$ | $5.510_{\pm 1.420}$ | $1.467_{\pm 12.346}$ |
| FNSDA | $\mathbf{2.555}_{\pm 1.330}$ | $\mathbf{0.109}_{\pm 0.168}$ | $7.533_{\pm 0.128}$ | $0.239_{\pm 0.079}$ | $\mathbf{2.746}_{\pm 0.995}$ | $2.252_{\pm 7.950}$ | $3.835_{\pm 1.167}$ | $0.741_{\pm 6.120}$ |

## C  FURTHER RESULTS AND ANALYSIS

### C.1  DETAILED RESULTS

In this section, we provide more detailed experimental results for our generalization tasks. The results on the inter-trajectory prediction task are presented in Table 4, and on the extra-trajectory prediction task are shown in Table 5. We further report in-domain test results in Table 6 to show the impact for the seen environments.

### C.2  INITIAL VALUE AND ENVIRONMENTAL PARAMETERS

To compare the discrepancies in terms of Fourier frequencies in a dynamical when initial conditions or PDE coefficient vary, we visualize their influence on generated trajectories by making a comparison to a fixed trajectory with $\nu = 8 \cdot 10^{-4}$ on the NS dataset. The results are depicted in Fig. 5. As seen, varying initial values can change the flow dynamic immensely, along with significant changes in low and high Fourier spectrums. While varying victory tends to shift the flow in nearby regions, and it can also change the low and high Fourier spectrum due to error accumulation. We, in our experiments, report the generalization results on different initial values and PDE coefficient simultaneously existing, which is a more challenging but realistic setup. To investigate the

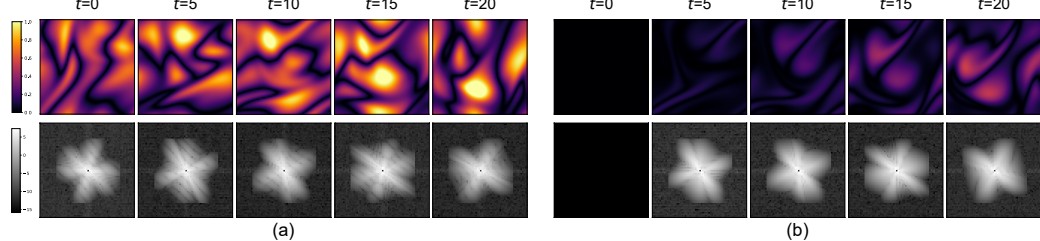

Figure 5: (a) Generated under different initialization; (b) Generated with $\nu = 1.1 \cdot 10^{-3}$.

effect of changing environmental parameters on the generated dynamics, we vary the parameter $\nu$ from $8 \cdot 10^{-4}$ to $\{9 \cdot 10^{-4}, 1.0 \cdot 10^{-3}, 1.1 \cdot 10^{-3}, 1.2 \cdot 10^{-3}\}$ and compare the resulting trajectories with the one obtained with $\nu = 8 \cdot 10^{-4}$ under the same initial value. The MSE and the Fourier representations of the differences are shown in Fig. 6. We can observe that the discrepancy between the trajectories increases as $\nu$ deviates from $8 \cdot 10^{-4}$, and this is reflected in both the low and high-frequency components of the Fourier domain. In addition, the discrepancy grows over time, indicating that the environmental parameter has a significant impact on the long-term dynamics.

### C.3  PARAMETER SENSITIVITY ANALYSIS

**The value of $\lambda$.** We constrain $c_e$ to be close to zero to facilitate fast adaptation to new environments. We then perform a parameter sensitivity analysis w.r.t. $\lambda$ on the LV dataset to assess its influence. The results are shown in Table 7. As seen, FNSDA exhibits stable performance under different strengths on the penalty of $c_e$.

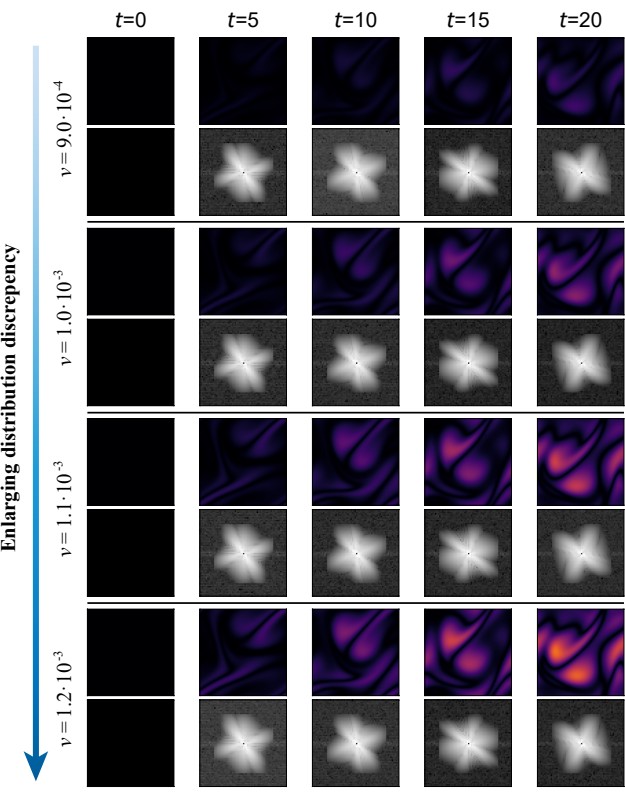

Figure 6: Comparison of different distribution discrepancies for the shift of dynamics. We visualize the differences between generated trajectories with $\nu \in \{9 \cdot 10^{-4}, 1.0 \cdot 10^{-3}, 1.1 \cdot 10^{-3}, 1.2 \cdot 10^{-3}\}$ and a trajectory that is obtained from the same initial value but a different value of $\nu = 8 \cdot 10^{-4}$.

Table 7: Parameter sensitivity analysis w.r.t $\lambda$ on LV dataset. We report the RMSE ($\times 10^{-2}$) results.

| $\lambda$ | 1e-3 | 1e-4 | 1e-5 | 1e-6 |
|---|---|---|---|---|
| Inter-trajectory | 4.631 | 3.736 | 3.783 | 4.705 |
| Extra-trajectory | 38.503 | 33.774 | 33.896 | 34.588 |

**The dimension of $c_e$.** We then analyze the parameter sensitivity with regard to the dimension of $c_e$ for our FNSDA by varying the dimension ranging from $\{2, 5, 10, 15, 20\}$. The results are listed in Table 8. FNSDA achieves the best performance when the dimension of $c_e$ is set to 10. It also shows consistent results for dim 2, 15, and 20, and slightly deteriorates for dim 5. We speculate that $c_e$, as a key component of FNSDA, learns to infer the latent code of the actual environmental parameters when generalizing to a new environment, thus its dimension is closely related to its learning ability.

Table 8: The effect of environmental parameter dimension on the LV dataset. We report the RMSE ($\times 10^{-2}$) results.

| $d_c$ | 2 | 5 | 10 | 15 | 20 |
|---|---|---|---|---|---|
| Inter-trajectory | 5.991 | 12.966 | 3.736 | 5.659 | 5.090 |
| Extra-trajectory | 34.751 | 42.632 | 33.774 | 50.981 | 48.927 |

**The impact of $\beta_e$.** We experimentally evaluate its impact with the results in RMSE presented in Table 9. We can observe that, fixing $\beta_e$ slightly worsens the performance, while fixing $c_e$ significantly degrades the performance. This demonstrates the critical role of $c_e$ for FNSDA's generalization capability.

Table 9: The effect of $\beta_e$ on the LV dataset. We report the RMSE $(\times 10^{-2})$ results.

| $d_c$ | 2 | 5 | 10 | 15 | 20 |
|---|---|---|---|---|---|
| Inter-trajectory | 5.991 | 12.966 | 3.736 | 5.659 | 5.090 |
| Extra-trajectory | 34.751 | 42.632 | 33.774 | 50.981 | 48.927 |

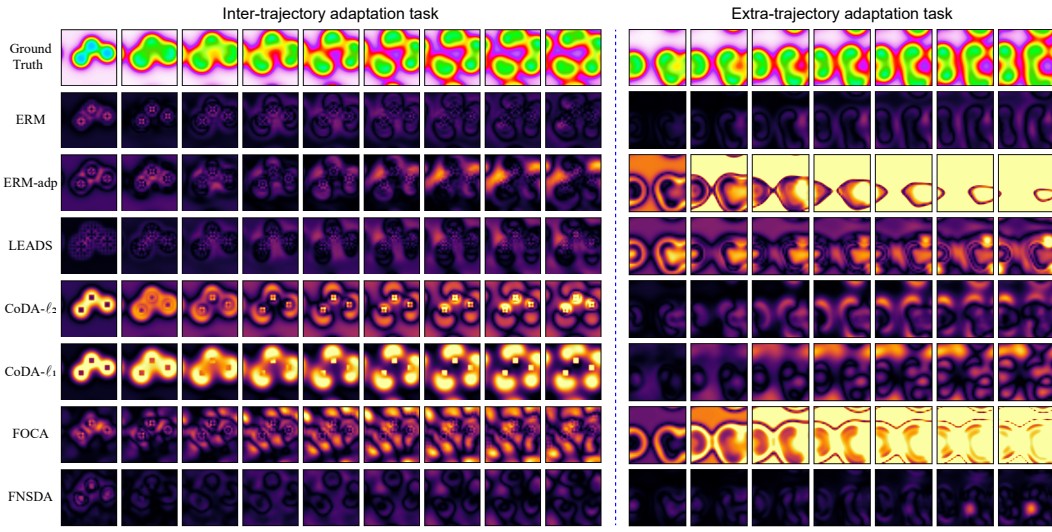

Figure 7: Adaptation results to new GS system with $(F, k, D_u, D_v) = (0.33, 0.61, 0.2097, 0.105)$. We present the ground-truth trajectory and prediction MSE per frame generated by different neural network simulators.

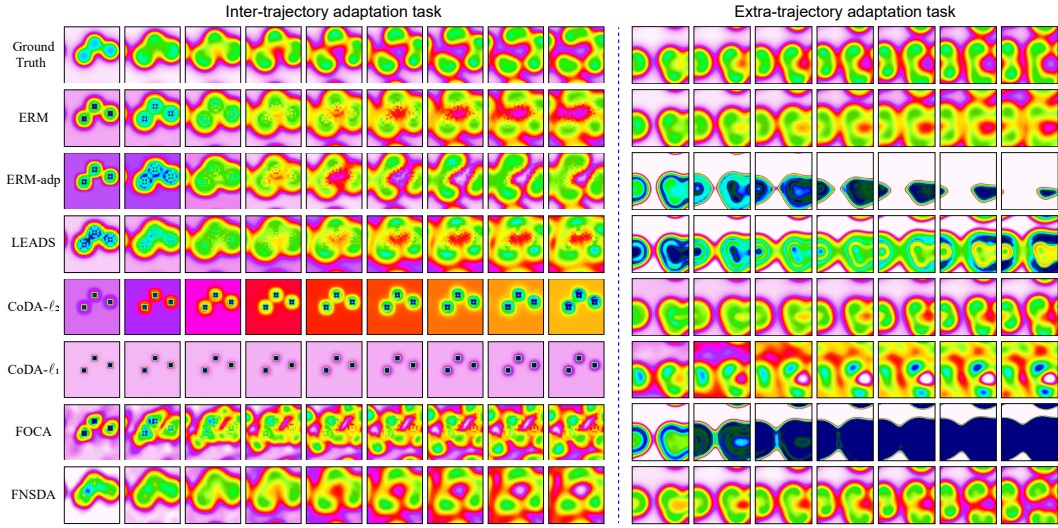

Figure 8: Visualization of predicted dynamics for a new GS system with $(F, k, D_u, D_v) = (0.33, 0.61, 0.2097, 0.105)$. We show the ground-truth trajectory and predictions from different neural network simulators.

## C.4 QUALITATIVE ANALYSIS

**Results on the GS dynamics.** We visualize in Fig. 7 prediction MSE by comparison method and our FNSDA for the inter-trajectory and extra-trajectory adaptation tasks on the GS dataset. The predicted dynamics are illustrated in Fig. 8.

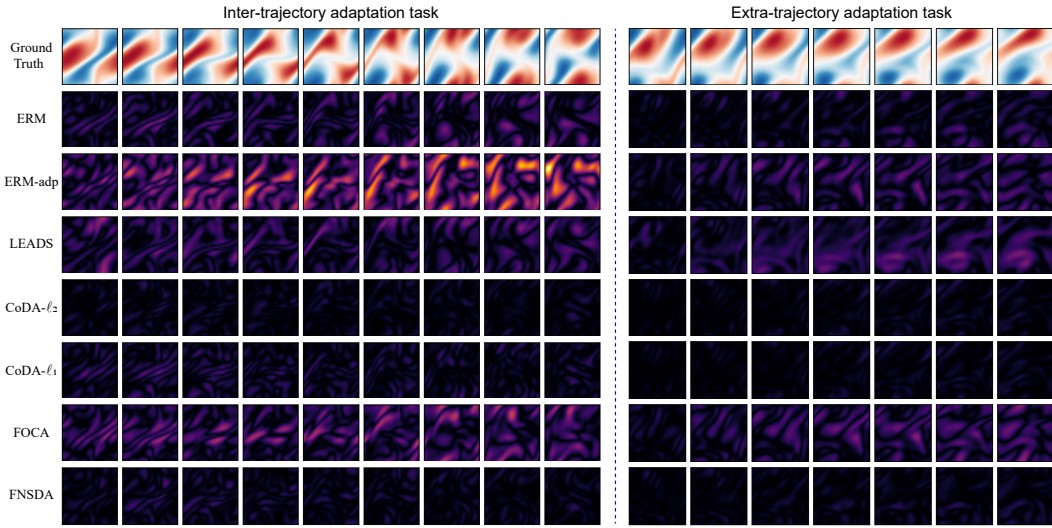

Figure 9: Adaptation results to new GS system with $\nu = 1.15 \cdot 10^{-3}$. We present the ground-truth trajectory and prediction MSE per frame generated by different neural network simulators.

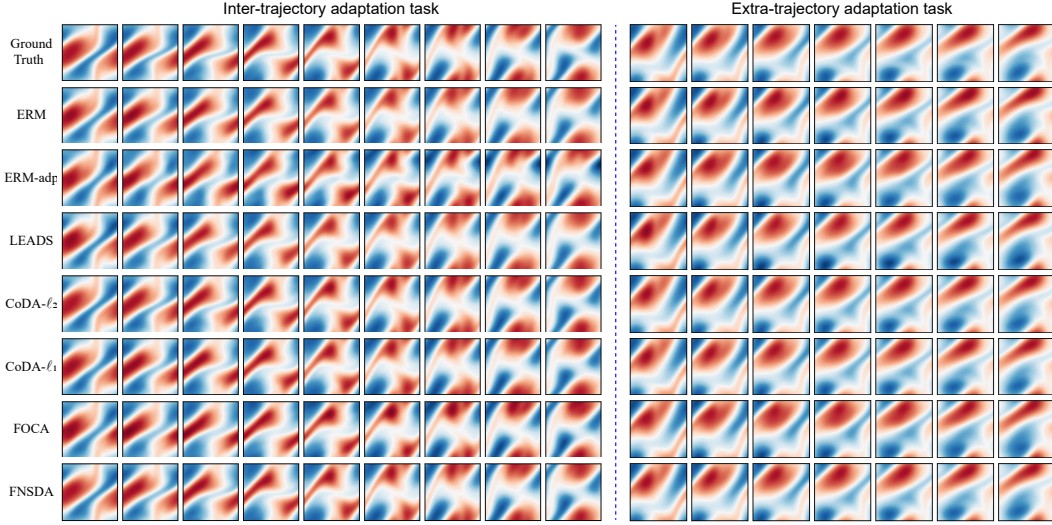

Figure 10: Visualization of predicted dynamics for a new NS system with $\nu = 1.15 \cdot 10^{-3}$. We show the ground-truth trajectory and predictions from different neural network simulators.

**Results on the NS dynamics.** We also visualize in Fig. 9 prediction MSE by comparison method and our FNSDA for the inter-trajectory and extra-trajectory adaptation tasks on the NS dataset. The predicted dynamics are illustrated in Fig. 10.

