# OpenReview forum: "Generalizing to New Dynamical Systems via Frequency Domain Adaptation"
_ICLR.cc/2024/Conference — Submitted to ICLR 2024_

### Official Review · Reviewer_nLXT · 2023-10-30

**Soundness:** 3 good
**Presentation:** 3 good
**Contribution:** 3 good
**Rating:** 6
**Confidence:** 4

**Summary:**

This paper addresses the generalization of neural surrogates of physics systems in forecasting heterogeneous dynamics. A novel method is proposed that 1) features on the frequency domain are used to generalize to unseen dynamics and 2) strategies such as separating commonalities and discrepancies among environments and swish activation are used in training the model. Empirical results show improved forecasting performance.

**Strengths:**

- The problem of adapting neural forecasting models on heterogeneous dynamics is important.
- Empirical results are obtained by comparison with benchmarks and tested on the chosen datasets.

**Weaknesses:**

- In Section 1 the author mentioned the limitations of meta-learning that they require updating a large number of parameters. This is true in optimization-based meta-learning, but not always true in other settings such as feed-forward-based meta-learning and metric-based meta-learning.
- The methodology is confusing. First, although the related work demonstrates the benefit of the Fourier transform in computation, the benefit of Fourier transform vs traditional methods such as recurrent neural networks and neural ODE-based methods in adaptation is missing. Second, the inconsistency of notations in paragraphs and figures makes it difficult to understand.
- The details of comparison baseline models need to be provided. No visual comparison of the prediction and the ground truths in experiments and ablation studies. The ablation study could be more complete.

**Questions:**

- The methodology is incremental based on CoDA (weights separated into environment-specific and environment-invariant parts) and using frequency domain features. The experimental results also show that CoDA has compatible performance with the proposed method on some of the datasets in both inter- and extra-trajectory adaptation. It would be more convincing if the authors add an ablation study on either using the frequency domain on CoDA or using features on temporal domains in the proposed method.
- Could the author address the comparison with feed-forward-based meta-learning? Feed-forward-based meta-learning does not require optimizing a large number of parameters and extra adaptation steps on test environments. It would be nice to mention this type of work, such as https://openreview.net/pdf?id=7C9aRX2nBf2, as related work.
- There is confusion in Fig 2 and Eq 4: if z^{(l)} is a time instant u(t) as described in Fig 2, what is the benefit of using the Fourier transform and inverse Fourier transform in comparison with traditional convolutional neural networks?
- Could the author explain the notations in methodology? For instance, what is F_e? What is the dimension of z^{(l)}? What are the dimension of W_{env}^{(l)}, f^e in Eq 8?
- Why is the ERM-adp worse than ERM in both settings? How is the adaptation performed on the model? Could the author provide more details of training the baseline models?
- The improvement by partition seems not significant compared to using all Fourier modes as shown in Fig 3. Is it specific to the dataset or a universal phenomenon among all datasets?
- The swish activation with the cosine annealing scheduler contributed significantly to the improvement of the performance. It would be fair to compare the results if all other baseline models were trained with the same activation function and scheduler.

---

> ### Author Response · Authors · 2023-11-22
> **Reply to reviewer nLXT**
>
> Thank the reviewer for the constructive feedback.
>
> **Distinction from CoDA:** We highlight two main contributions distinguishing our work from CoDA: (1) we are the first to emphasize adaptation in the frequency domain, and demonstrate its potential for generalizing to unseen environments. (2) Our hierarchical disentangling via the splitter $K$ and rapid adaptation by conditioning on $c_e$ results in memory and computational efficiency, facilitating its practical use. For experiments on NS dataset, CoDA uses FNO as the backbone, while it needs to update much more parameters (0.930M) than our approach (0.096K). Moreover, CoDA requires more optimization iterations for convergence as observed in Figure 4 (c)&(d).
>
> **Comparison with feed-forward-based meta-learning:** We agree that the presented feed-forward-based meta-learning approach doesn't need to update parameters during test, it adjusts the predictor implicitly using few-shot observations. We have included this paper in revised version. Here, we evaluate three Bayesian meta-learning methods introduced in this reference for inter-trajectory adaptation tasks on NS dataset:
> |Model|RMSE$(\times 10^2)$|MAPE|
> |:---:|:---:|:---:|
> |meta-GRU-res|61.817±5.427|0.786±0.065|
> |meta-NODE|62.224±5.506|0.852±0.435|
> |meta-RGN-res|62.014±5.479|0.812±0.129|
> |FNSDA (ours)|3.625±0.882|0.355±0.579|
>
> As seen, these considered methods cannot produce accurate predictions for unseen dynamics. We conjecture that this is mainly due to two factors: (1) heavy reliance on a large meta-training set with diverse dynamics. In their original experimental setup, there are 16 environments, each equipped with 4096 trajectories for training, while we only have 5 environments with 50 trajectories each. (2) Too scarce observation for adaptation. They require 3-shot trajectories and an observed window of 5 for adaptation, while we are in a 1-shot task and the observed window is 1. As a result, their inferred initial value for the latent dynamics is unreliable.
>
> **Benefits of Fourier transform-based network over CNNs:** CNNs are known to have a ``spectral bias’’ that favors low-frequency functions and limits their ability to represent the high-frequency content [Rahaman2019] which is crucial for accurate dynamics forecasting. On the other hand, Fourier transform-based networks can easily capture the higher frequency information [Tancik2020] and have demonstrated their capability to model complex PDEs [Li2021]. Furthermore, each FNO layer processes global and local information simultaneously via weight multiplication of different modes, so it can naturally incorporate our hierarchical disentangling via the splitter $K$ and rapid adaptation via conditioning on $c_e$, resulting in improved memory and computational efficiency for adaptation.
>
> **Clarifications of notations:** $F_e$ is the vector field describing the ground truth dynamics, $f^e$ in Eq. 8 is the same as $F_e$. $z^{(l)} \in \mathbb{R}^{m}$ and $W_{env}^{(l)} \in \mathbb{C}^{\hat{k} \times m \times d_c}$. We have carefully revised the notations in our manuscript for clarity.
>
> **ERM-adp performs worse than ERM:** ERM-adp updates all model parameters without regularization, which makes it susceptible to overfitting the observations and leads to inferior outcomes than ERM. The adaptation is performed by minimizing the loss on observations using Adam.
>
> **Comparison with using all Fourier modes:** FNSD achieved only a marginal improvement over using all Fourier modes mainly attributed to the already high prediction precision, as well as the fact that the modes acquired by Fourier transform are limited for LV dataset, our model tends to employ more modes for better performance.
>
> **Using swish activation and cosine annealing schedule:** In our original experiments, we kept all other settings the same for the comparison baselines, including using swish activation in backbone networks and Adam optimizer. We evaluate them with/without cosine annealing schedule on LV dataset:
>
> ||No annealing schedule||+ Annealing schedule||
> |:---:|:---:|:---:|:---:|:---:|
> ||Inter-trajectory|Extra-trajectory|Inter-trajectory|Extra-trajectory|
> |ERM|48.310|43.969|48.306|48.306|
> |ERM-adp|47.284|95.193|47.071|95.173|
> |LEADS|69.604|88.214|79.588|93.575|
> |CoDA-$\ell_2$|4.674|29.660|4.480|29.643|
> |CoDA-$\ell_1$|5.044|31.088|5.462|33.038|
> |FOCA|21.321|77.046|27.403|77.816|
>
> Cosine annealing schedule had a negative impact on most of the comparison methods, indicating it is not a universal technique that can benefit all methods.
>
> We have revised our manuscript accordingly and provided qualitative comparisons in the Appendix.
>
> **References**
>
> [Rahaman2019] Nasim Rahaman, et al. On the Spectral Bias of Neural Networks. ICML, 2019.
>
> [Tancik2020] Matthew Tancik, et al. Fourier Features Let Networks Learn High Frequency Functions in Low Dimensional Domains. NeurIPS, 2020.
>
> [Li2021] Zongyi Li, et al. Fourier Neural Operator for Parametric Partial Differential Equations. ICLR, 2021.

---

> > ### Comment · Reviewer_nLXT · 2023-11-23
> >
> > Thank the authors for the clarification and additional experiments. I modified my evaluation accordingly.

---

### Official Review · Reviewer_YVis · 2023-10-31

**Soundness:** 2 fair
**Presentation:** 2 fair
**Contribution:** 3 good
**Rating:** 6
**Confidence:** 3

**Summary:**

This paper tackles the generalization to new physical systems of DL architectures. The architecture consists of a modified version of a Fourier Neural Operator (FNO). Adaptation is performed in the frequency domain with a parameter $c_e$ and in the projected space with the parameter $\beta_e$ of the Swish activation function, which changes for each layer. In the frequency domain, frequencies are decomposed into two components that are added, a constant one across environments and an environment-specific one. The constant component consists of the multiplication of the Fourier transform with two learned matrices, $R_s^{(l)}$ and $(1-K)$ and the environment-specific one consists of the same multiplication with two matrices, $W_{env}^{(l)} c_e$ and $K$. The filter $K$ then acts as a selection of the different frequencies that are split into the two groups. The loss is an integral of a MSE over the temporal domain, thus needing numerical solvers to compute it. Adaptation is performed by updating only $c_e$ and $(\beta_e^{(l)})_l$ once the model is trained. The architecture is then tested on two settings, a one-shot adaptation and an unseen future prediction, using four different datasets.

**Strengths:**

- Interesting to perform adaptation mostly in the frequency domain.
- The method seems to be well-trained and details on the influence of the different training improvements are presented.
- Quantitative results are promising.

**Weaknesses:**

- The modification of the original FNO architecture is not very important.
- There is no qualitative results presented (except Figure 1 which is not really commented). This is then hard to really capture if the model is able to generalize well.
- It would be interesting to see the adaptation of the method with an increasing number of trajectories to adapt from. Currently, only-one shot adaptation is performed.
- Adaptation is performed by updating $c_e$, it would be important to have an ablation study on the influence of the dimension of $c_e$ on the performances.
- An interesting ablation study would be to make the $\beta_e$ of the Swish activation function a constant. Indeed, in the current architecture, the adaptation is not completely done in the frequency domain, thus it is hard to understand the influence of this adaptation. A follow-up minor concern would be that only updating $\beta_e$ at adaptation would lead to similar results to the current architecture.
- This is minor but the ordinate of Figure 4 c) and d) should be in log-scale, it is impossible to distinguish the middle and end of the training curve.

**Questions:**

- Have you studied the influence of the number of modes of the architecture for the different datasets?
- Why did you chose Euler solver to compute the loss for the NS dataset and RK4 for the other datasets?
- How did you chose $T_{ad}$ and $T$ for the extra-trajectory adaptation? It seems to depend on the discretization which is coherent, but I wonder how would changing $T_{ad}$ and $T$ would change the results. Have you tried with different values?
- Have you compared the adaptation times of the different methods?

---

> ### Author Response · Authors · 2023-11-22
> **Reply to reviewer YVis**
>
> Thank the reviewer for the valuable questions.
>
> **Modification of FNO:** We highlight our three main contributions in this submission as follows: (1) We address the generalization issue of neural learned simulators which is an essential but rather new area. The original FNOs suffer from poor generalization when facing distribution discrepancy. (2) We propose a novel hierarchical disentangling and rapid adaptation mechanism in the Fourier domain, which is lightweight, memory-efficient and computationally fast. (3) We experimentally demonstrate the superior generalization capability of our approach for various dynamical systems in a parameter-efficient and adaptation-efficient manner.
>
> **Qualitative results:** Thanks for the suggestion, we have included qualitative comparisons on the GS dataset (Figure 7 and 8 in Appendix C) and NS dataset  (Figure 9 and 10 in Appendix C) in our revised manuscript.
>
> **Adapting using an increasing number of trajectories:** We experimentally compare the performance of CoDA with l2 regularization and our FNSDA with different numbers of trajectories for the inter-trajectory adaptation task on the LV dataset. The results are presented in the table below:
> |$N_{adp}$|1|5|10|20|50|
> |:---:|:---:|:---:|:---:|:---:|:---:|
> |CoDA-$\ell_2$|4.674|4.672|4.708|4.653|4.646|
> |FNSDA|3.736|3.402|3.298|3.200|3.204|
>
> FNSDA achieves a consistent decrease in error as the number of trajectories for adaptation increases. This indicates that FNSDA can effectively leverage more data to improve its generalization ability.
>
> **The dimension of $c_e$:** We examine the performance of FNSDA with different dimensions of $c_e$ ranging from {2, 5, 10, 15, 20}. The results in RMSE are shown in the table below:
> |dim($c_e$)|2|5|10|15|20|
> |:---:|:---:|:---:|:---:|:---:|:---:|
> |Inter-trajectory|5.991|12.966|3.736|5.659|5.090|
> |Extra-trajectory|34.751|42.632|33.774|50.981|48.927|
>
> FNSDA achieves the best performance when the dimension of $c_e$ is set to 10. It also shows consistent results for dim 2, 15, and 20, and slightly deteriorates for dim 5. We speculate that $c_e$, as a key component of FNSDA, learns to infer the latent code of the actual environmental parameters when generalizing to a new environment, thus its dimension is closely related to its learning ability.
>
> **Ablation on $\beta_e$:** We experimentally evaluate its impact with the results in RMSE presented as follows
> |   |Fixing $\beta_e$|Fixing $c_e$|Update both|
> |:---:|:---:|:---:|:---:|
> |Inter-trajectory|5.991|12.966|3.736|
> |Extra-trajectory|34.751|42.632|33.774|
>
> As seen, fixing $\beta_e$ slightly worsens the performance, while fixing $c_e$ significantly degrades the performance. This demonstrates the critical role of $c_e$ for FNSDA’s generalization capability.
>
> **Fig. 4 (c) and (d):** Thanks for the suggestion, we have improved this figure for better visual clarity and presentation.
>
> **The influence of the number of modes:** In our preliminary experiments, we found that reducing the number of Fourier modes would consistently lead to worse performance. This is because this operation impairs the recovery of input signals after performing inverse FFT. This motivates our introduction of a learnable splitter $K$ that allows us to preserve all Fourier modes but only update some environment-specific part of them during adaptation.
>
> **The choice of solver:** We chose the solver mainly for the sake of fairness. We followed the same solver selection as the official implementations of [Yin2021] and [ Kirchmeyer2022].
>
> **The effect of $T_{ad}$:** We fix the total length as $T=20s$ and vary the value of $T_{ad}$ ranging from {2.5, 5, 7.5, 10, 14} for extra-trajectory adaptation task on the LV dataset, the results are listed below
> |$T_{ad}$|2.5s|5s|7.5s|10s|15s|
> |:---:|:---:|:---:|:---:|:---:|:---:|
> |FNSDA|45.391|34.751|19.355|17.367|8.776|
>
> As seen, increasing the value of $T_{ad}$​ leads to a consistent improvement in prediction accuracy, which aligns with our intuition on the matter.
>
> **Time cost for adaptation:** We measure the adaptation time cost by reporting the number of iterations required for convergence in Figure 4 (c) and (d). Each method takes a similar amount of time for one iteration, except for FOCA, which employs a meta-learning framework involving bi-level optimization in which each iteration consists of 10 gradient descent steps in the inner loop.
>
>
> **References**
>
> [Yin2021] Yuan Yin, Ibrahim Ayed, Emmanuel de Bézenac, Nicolas Baskiotis, Patrick Gallinari. LEADS: Learning Dynamical Systems that Generalize Across Environments. NeurIPS, 2021.
>
> [Kirchmeyer2022] Matthieu Kirchmeyer, Yuan Yin, Jeremie Dona, Nicolas Baskiotis, Alain Rakotomamonjy, Patrick Gallinari. Generalizing to New Physical Systems via Context-Informed Dynamics Model. ICML, 2022.

---

> > ### Comment · Reviewer_YVis · 2023-11-23
> > **Thank you for your answers and additional experiments**
> >
> > Thank you for your answers and additional experiments. Some of my main concerns have been addressed, especially on the architectures choices. I find the ablation on $\beta_e$ particularly interesting. The qualitative results also help in better understanding the performances. I thus raise my score.

---

### Official Review · Reviewer_fQXG · 2023-10-31

**Soundness:** 3 good
**Presentation:** 3 good
**Contribution:** 2 fair
**Rating:** 5
**Confidence:** 5

**Summary:**

This paper presents a method to improve the generalization of trained DNN model for dynamic systems. The method is based on FNO. The paper presents the network design, theory, and some empirical validation of the proposed method.

**Strengths:**

The motivation of the proposed method is sound, and the research efforts in this direction can represent significant advancement of DL for dynamic system behavior modeling. The idea presented in the manuscript is novel, but intuitively makes sense. The paper does a good job explaining the ideas. The experimental results do provide some validation on the effectiveness of the proposed method.

**Weaknesses:**

The narrative of the paper can be further improved. The experimental results also can be further improved by more comprehensive test cases. See the questions section for more details.

**Questions:**

1. My biggest question is the meaning of `environment` in the context. For an ODE, one would need the model coefficients and the initial conditions (IC). For a PDE, one would have to add the boundary conditions (BC). Since the method is built on FNO, which should be effective to cover different ICs. The main text didn't explain this part well. Based on my reading of the supplemental material, seems by `environment`, the authors mean model coefficients of different values.

2. Following up to the previous point, a better narrative is to use uncertainties: `aleatoric` and `epistemic`.

3. Two of the experiments are ODEs, which are not the strong points of FNOs. I would like to see more experiments on various types of PDEs

4. From supplemental materials, the `deviations` from the model coefficients used for training and to adaptation seem to nicely covered. One example is the Glycolitic oscilators, the training $k_1 \in \{100, 90, 80\}$, and the evaluation of $k_1$ is bracket value of $85$ and $95$.  What happens when $k_1$ values chosen far away from the training interval, e.g., $125$. Some ablation study to show when the method will fail can actually give the readers a better understanding of the proposed method.

5. For NS example, what are the equivalent Reynolds number in different training and evaluation settings?

---

> ### Author Response · Authors · 2023-11-22
> **Reply to reviewer fQXG**
>
> Thank the reviewer for recognizing our contribution and offering helpful writing suggestions.
>
> **The definition of `environment`:** In our manuscript, we use the term ‘environment’ to refer to the specific vector field $F_e$ that governs the dynamics of the system. Trajectories that are generated from different initial values but the same $F_e$  are regarded as belonging to the same environment. We acknowledge that varying the boundary conditions (BC) can also produce different environments, but we follow the previous works  [Yin2021] and [Kirchmeyer2022] that only investigate the generalization across different model coefficients in $F_e$. We visualize their impact on the flow dynamics by changing the viscosity in Figure 6 in the Appendix. As shown, varying the viscosity tends to shift the flow in nearby regions, but it can affect the low and high Fourier spectrum due to error accumulation. We leave the exploration of the effect of BC as future work.
>
>
> **More experiments on PDEs:** We agree with the importance of exploring more PDEs. In this submission, we follow the setting of [Kirchmeyer2022] and [Yin2021] to explore the generalization issue for generic neural learned simulators. PDEs, including the Glycolitic-Oscillator and Navier-Stokes dynamics, are considered. We would like to claim that current research on neural learned simulators mainly focuses on the approximation capability for complex physical systems. The generalization issue, although crucial, is still in its infancy. We believe our work provides a valuable foundation for inspiring future research on neural learned simulators, especially for their practical applications.
>
>
> **Deviations in the model coefficients:** This is an interesting experiment and we have conducted a similar setup on the LV dataset with the results shown in Fig 4. (b). In this experiment, the training environments are set as $\beta, \delta \in \\{0.5, 0.75, 1.0 \\}^2$, the *interpretable* test environment uses $\beta=0.625, \delta=0.625$ and the *not interplatable* test environment uses $\beta=1.125, \delta=1.125$. As observed, all methods perform worse in the *not interpretable* test environment, and our method consistently achieves a low forecast error in this challenging scenario.
>
>
> **Reynolds number for NS dataset:** The Reynolds numbers for the five training environments are given by $\\{1250, 1111, 1000, 909, 833\\}$ and for the four test environments by $\\{1176, 1053, 952, 870\\}$.
>
>
> **References**
>
> [Kirchmeyer2022] Matthieu Kirchmeyer, Yuan Yin, Jeremie Dona, Nicolas Baskiotis, Alain Rakotomamonjy, Patrick Gallinari. Generalizing to New Physical Systems via Context-Informed Dynamics Model. ICML, 2022.
>
>
> [Yin2021] Yuan Yin, Ibrahim Ayed, Emmanuel de Bézenac, Nicolas Baskiotis, Patrick Gallinari. LEADS: Learning Dynamical Systems that Generalize Across Environments. NeurIPS, 2021.

---

### Official Review · Reviewer_uCSb · 2023-11-02

**Soundness:** 3 good
**Presentation:** 3 good
**Contribution:** 2 fair
**Rating:** 6
**Confidence:** 4

**Summary:**

This work proposes an adaptation to the traditional Fourier neural operator (FNO) to learn on trajectories that follow the same differential equations (with parameters that might change. Trajectories from a single set of parameters is referred to as an environment).

The core idea is that frequencies define the trajectory observed in an environment.  In that sense, the authors propose to model the relevant frequencies as an additive combination of “common frequencies” and “specific frequencies”. In line with this remark, the authors then propose an environment specific combination of the fourier modes. This provides the authors with a simple framework to test their method on a variety of DE for both in domain prediction and out of domain adaptation.

The overall algorithm is learned to minimize the prediction error on several time steps and additional penalty on the environment specific variable $c_e$ that conditions the env-specific FNO layer.

**Strengths:**

Overall, the paper is fairly well written and relevant baselines are selected in the experimental section (which is fairly well conducted).

It builds up upon FNO paper and  (Kirchmeyer et al). Note that the proposition is conceptually close to the latter, adapting the model in the frequency domain whereas (Kirchmeyer et al) proposes an adaptation directly in the neural network parameters domain.

**Weaknesses:**

1.  I would have enjoy at least experimental considerations showing the discrepancies in terms of Fourier frequencies in a dynamical when initial conditions or PDE coefficient vary.

2. Note that i believe that this can also be done more theoretically analysing the Fourier modes of some simple equations such as the wave equation. Such a work could be a nice motivation for the proposed approach

3. What are the main limitation of the work ? Do you know about systems that behave "no continuously" in the frequency domain, e.g. that do not conserve frequencies when varying some condition ?

4. I find Fig.4 difficult to read.

**Questions:**

Questions:

1. Should not $K$ (eq.5) depend on the environment itself ? Indeed, it seems natural that the splitting in the frequency domain depends on the environment ?
2. Can the authors provide an intuitive explanation for $W_e$ and $W_s$ ?
3. Can the authors detail the meaning of penalizing $c_e$ ? What is the influence of the value of $\lambda$ on training ?
4. It would have been nice to compare the data intensiveness of the proposed algorithm. For instance comparing the prediction/adaptation error for selected models with the number of trajectories available during training.

---

> ### Author Response · Authors · 2023-11-22
> **Reply to reviewer uCSb**
>
> Thank the reviewer for the detailed review and constructive comments.
>
> **Discrepancies in Fourier space when initial conditions or PDE coefficient vary:** We agree with the reviewer’s suggestion that an experimental comparison of varying initial values and PDE coefficient for dynamics in the Fourier space can further enhance the clarity of our manuscript. To this end, we visualize their influence on generated trajectories by comparing their difference to a  fixed trajectory on the NS dataset (Figure 5 in Appendix C). As seen, varying initial values only can change the flow dynamic immensely, along with significant changes in low and high Fourier spectrums. While varying victory tends to shift the flow in nearby regions, and it can also change the low and high Fourier spectrum due to error accumulation. In our original submission, we report the generalization results on different initial values and PDE coefficient simultaneously existing, which is a more challenging but realistic setup. The generalization results on varying initial values only (in-domain test) have been included in our revised version in Table 6 of Appendix C.
>
> **Theoretical analysis of Fourier modes for differential equations:** We appreciate the reviewer’s suggestion that theoretical analyses on some simple differential systems can offer new insights for supporting our method. Generally, our work is theoretically motivated by that modeling in the Fourier domain can represent both high and low frequency content without the spectral bias of CNNs towards low frequencies [Rahaman2019]. We verify in Figure 5 that varying the environmental parameters can influence the dynamics both in low and high frequencies.  Besides, these low and high frequencies, capturing the local and global information of input signals, can be separated by a customized filter [Tveito1998]. This facilitates our learnable splitter to disentangle them. Last but not least, our trainable weight tensors are direct parameterizations of one kernel [Li2021], and we can modify them to tune the kernel’s spectrum as suggested by [Tancik2020].
>
> **Limitations of FNSDA:** As a neural network-learned simulator, FNSDA requires high-quality data for model training. Besides, its scalability towards much larger and more complex systems, such as those encountered in climate modeling or large-scale biological networks, is still underexplored. We discuss this in Appendix A in our revised version.
>
> **About Fig. 4:** The four subfigures show the ablation of training techniques, generalization to different distribution discrepancies, and convergence speed on two adaptation tasks, respectively. We have polished them for better visual clarity and presentation.
>
> **Learning the splitting via K:** The splitting is learned by minimizing the prediction error on the training environments, so it implicitly conditions on all the training environments.
>
> **Intuitive explanations for $W_e$ and $ W_s$:** These weight tensors are designed to modify the frequency spectra. They are direct parameterizations for the Fourier transform of a kernel function [Li2021].
>
> **The penalty on $c_e$:** We constrain $c_e$​ to be close to zero to facilitate fast adaptation to new environments. We then conduct a parameter sensitivity analysis w.r.t. $\lambda$ on the LV dataset to assess its influence. The results are shown in the table below:
> |$\lambda$|1e-3|1e-4|1e-5|1e-6|
> |:---:|:----:|:-----:|:----:|:----:|
> |Inter-trajectory|4.631|3.736|3.783|4.705|
> |Extra-trajectory|38.503|33.774|33.896|34.588|
>
> The results indicate that FNSDA has stable performance under different levels of penalty on $c_e$.
>
> **Data intensiveness analysis:** We empirically evaluate the performance of FNSDA with different numbers of trajectories for adapting to new environments. The results are presented in the table below:
> |$N_{adp}$|1|5|10|20|50|
> |:---:|:----:|:-----:|:----:|:----:|:----:|
> |CoDA-$\ell_2$|4.674|4.672|4.708|4.653|4.646|
> |FNSDA|3.736|3.402|3.298|3.200|3.204|
>
> The table shows that FNSDA achieves a consistent decrease in error as the number of trajectories for adaptation increases. This indicates that FNSDA can effectively leverage more data to improve its generalization ability.
>
> We have revised our manuscript according to the above discussion and provided qualitative comparisons in the Appendix to demonstrate the effectiveness of our method.
>
> **References**
>
> [Rahaman2019] Nasim Rahaman, et al. On the Spectral Bias of Neural Networks. ICML, 2019.
>
> [Tveito1998] Aslak Tveito, et al. Introduction to Partial Differential Equations: A Computational Approach. Springer, New York, 1998.
>
> [Li2021] Zongyi Li, et al. Fourier Neural Operator for Parametric Partial Differential Equations. ICLR, 2021.
>
> [Tancik2020] Matthew Tancik, et al. Fourier Features Let Networks Learn High Frequency Functions in Low Dimensional Domains. NeurIPS, 2020.

---

### Meta-Review · Area_Chair_zJ3h · 2023-12-06

**Metareview:**

This paper proposes a novel adaptation of the traditional Fourier neural operator (FNO) for learning trajectories in different dynamic systems. Although the paper is well-written, with a sound theoretical basis and promising empirical results​​, there are significant concerns that need to be addressed. Reviewers pointed out the need for clearer experimental demonstrations, particularly regarding the discrepancies in Fourier frequencies under varying initial conditions or PDE coefficients, and a lack of theoretical analysis of Fourier modes for different equations​​. The authors responded by providing additional experimental results and theoretical insights in their rebuttal, addressing the concerns raised by the reviewers to some extent​​. Despite these efforts, the reviewers raised several issues that the authors' responses did not fully address. These include concerns about the method's generalizability, the practicality of its application, and a need for more comprehensive testing and validation. In conclusion, while the paper makes a novel contribution to the field and the authors have made efforts to address the reviewers' concerns, there remain significant gaps in the experimental validation and theoretical underpinning of the proposed method. The reviewers' responses, though appreciative of the authors' efforts, also indicate that the paper still lacks sufficient clarity and depth in its methodology and results. Therefore, considering the overall feedback from the reviewers and the authors' responses, I recommend rejecting this submission.

**Justification For Why Not Higher Score:**

While the paper makes a novel contribution to the field and the authors have made efforts to address the reviewers' concerns, there remain significant gaps in the experimental validation and theoretical underpinning of the proposed method. The reviewers' responses, though appreciative of the authors' efforts, also indicate that the paper still lacks sufficient clarity and depth in its methodology and results.

**Justification For Why Not Lower Score:**

N/A

---

### Decision · Program_Chairs · 2024-01-16

Reject